# Learning Goal-Conditioned Representations for Language Reward Models

**Vaskar Nath**[*][†]     **Dylan Slack**[*]     **Jeff Da**

**Yuntao Ma**     **Hugh Zhang**     **Spencer Whitehead**[‡]     **Sean Hendryx**[‡]

Scale AI

## Abstract

Techniques that learn improved representations via offline data or self-supervised objectives have shown impressive results in traditional reinforcement learning. Nevertheless, it is unclear how improved representation learning can benefit reinforcement learning from human feedback on language models. In this work, we propose training reward models (RMs) in a contrastive, *goal-conditioned* fashion by increasing the representation similarity of future states along sampled preferred trajectories and decreasing the similarity along randomly sampled dispreferred trajectories. This objective significantly improves reward model performance by up to 0.09 AUROC across challenging benchmarks, such as MATH and GSM8k. These findings extend to general alignment as well – on the Helpful-Harmless dataset, we observe 2.3% increase in accuracy. Beyond improving reward model performance, we show this way of training RM representations enables improved *steerability* because it allows us to evaluate the likelihood of an action achieving a particular goal-state (*e.g.,* whether a solution is correct or helpful). Leveraging this insight, we find that we can filter up to 55% of generated tokens during majority voting by discarding trajectories likely to end up in an "incorrect" state, which leads to significant cost savings. We additionally find that these representations can perform fine-grained control by conditioning on desired future goal-states. For example, we show that steering a Llama 3 model towards helpful generations with our approach improves helpfulness by 9.6% over a supervised-fine-tuning trained baseline. Similarly, steering the model towards complex generations improves complexity by 21.6% over the baseline. Overall, we find that training RMs in this contrastive, goal-conditioned fashion significantly improves performance and enables model steerability. [1]

## 1 Introduction

Aligning Language Models (LMs) with human preferences has proven to be an essential step for the adoption and safe use of these models, with the dominant paradigm being Reinforcement Learning from Human Feedback (RLHF) [50]. To accomplish this, a standard setup is to collect labels from humans for generated responses (*e.g.,* preferences, quality ratings) [50]. These labels can then be used to train a reward model to produce a ranking/scoring of a given sequence or set of sequences.

---

[*]Equal contribution

[†]Corresponding author: `vaskar.nath@scale.com`

[‡]Equal senior authorship

[1]Code available at `https://github.com/vaskarnathscale/goal-conditioned-rm`

38th Conference on Neural Information Processing Systems (NeurIPS 2024).

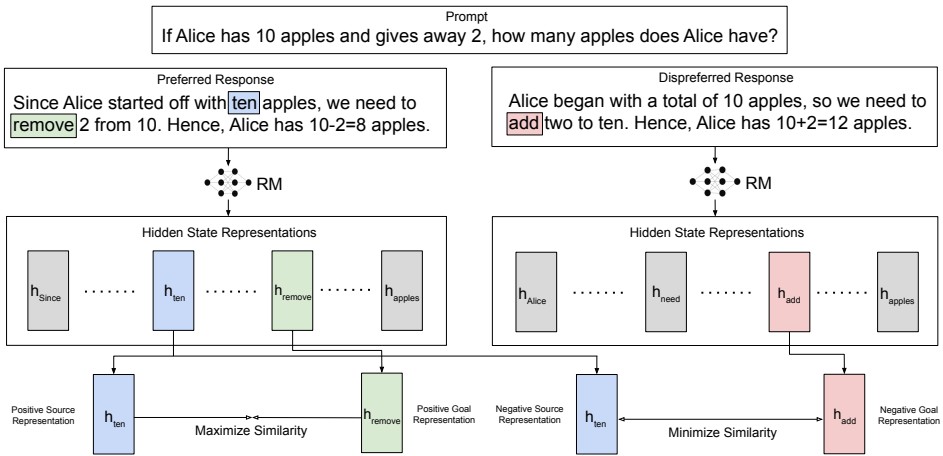

Figure 1: **Overview of contrastive goal-conditioned learning for text.** Pictured is a prompt with a preferred and dispreferred response. Both source state tokens (ten) for the positive and negative trajectory are sampled from the preferred response. For illustrative purposes, the positve and negative source states are sampled as the same token, but in practice they can be different. The positive goal state is sampled as some future token (subtract) from the preferred response, and the negative goal state is sampled from any token (add) from the dispreferred response. The corresponding representations are retrieved from the last hidden state of the reward model. The training objective is then to maximize and minimize the similarity of the positive and negative representation pairs, respectively.

The policy LM is then trained to maximize the expected returns from this reward model using a Reinforcement Learning (RL) algorithm.

High-quality representations have been shown to be an important piece for the success of RL algorithms [6, 42]. Although such representations can be learned during end-to-end training, many efforts have found it important to integrate more explicit representation learning components into RL algorithms, such as via data augmentation [39] or auxiliary losses [21]. Some work even casts certain RL algorithms as representation learning methods where using the similarity between state representations serves as a value function, demonstrating success on manipulation and navigation tasks [20]. Despite these successes in different areas, representation learning for aligning LMs has been less explored, while more emphasis has been placed on, e.g., pre-training reward models [7, 37] or learning from different types of rewards [15, 38].

In this paper, we present a simple yet effective approach to improve the representations learned by reward models for aligning LMs. We train LM-based reward models to learn representations that capture the expected reward or likelihood of achieving a goal state (*e.g.,* correct solution to a problem, helpful response) at intermediate steps of the input sequence, inspired by goal-conditioned RL [5, 13, 20]. To do so, we use a contrastive objective and apply it to the reward model's hidden representations from desirable and undesirable sequences. Enforcing this loss on representations from intermediate steps of the sequence helps encode a dense signal as to which trajectories are more promising at different points in the sequence, which we show offers several useful properties, such as helping to localize errors or evaluating partial completed sequences. This method is flexible enough to support different kinds of alignment data and does not require further annotations beyond common sequence-level annotations.

We find that this approach improves the reward model's ability to identify correct/incorrect solutions in mathematical reasoning, boosting the AUROC on the task by up to 0.09 over standard preference ranking training. Towards natural language alignment, we find this method is able to increase the reward model's accuracy of identifying helpful versus harmless responses by 2.3% on the Helpful-Harmless dataset [8].

We also demonstrate the utility of the learned representations themselves, e.g., for filtering solutions to improve accuracy and steering the outputs towards responses with certain attributes in guided decoding. On mathematical reasoning, we show that we are able to filter up to 55% of generated

tokens by discarding trajectories that are likely to lead to incorrect solutions as deemed by the learned representations while achieving similar or better performance. Similarly, using these representations to steer a Llama 3 [1] model by conditioning on desired future goal-states, we improve helpfulness by 9.6%, correctness by 12.2%, coherence by 16.5%, and complexity by 21.6% over the supervised-fine-tuning trained baseline.

In summary, our contributions are as follows: 1) We explore improving the learned representations of reward models and its effect on LM alignment. Towards this, we present a simple and effective representation learning method based on a goal-conditioned contrastive objective. 2) We demonstrate that training reward models with this method can improve reward model performance on mathematical reasoning and helpfulness/harmlessness benchmarks. 3) We show that simply utilizing a reward model trained with this method can improve policy LM alignment on math reasoning benchmarks. 4) We investigate using these representations as a mechanism to evaluate the likelihood of achieving a desired goal state by filtering generations in a majority-vote scheme and guided decoding, showing that they can be used to increase accuracy and control policy LM outputs.

## 2 Related Work

**Language model (LM) alignment and Reinforcement Learning from Human Feedback (RLHF).** The goal of RLHF [50, 71, 76] for aligning LMs is to train a policy LM that generates responses that are preferred by humans. This framework has proven instrumental for increasing the utility and generalization of LMs, improving their capabilities on a wide variety of tasks, and their safety [8, 9, 45, 61, 49]. Critically, RLHF improves the policy LM along a scalar *reward* signal, which indicates the value of a generated piece of text, and their efforts have focused on different means of providing such signals [7, 56]. Typically, researchers train a *reward model* that learns to score generations based on pair-wise human preference data. Subsequently, the model provides reward signals during reinfocement learning (RL) training of the policy, often via Proximal Policy Optimization (PPO) [59]. While reward models have often been trained to score text with just a single scalar value, several works have shown that training the reward model to provide fine-grained reward signals over *spans* of text can help improve model performance by providing intermediate rewards for the policy LM to learn from during RL [43, 68]. Nevertheless, fine-grained reward methods often require extensive and costly human annotations on the sentence or token level to train the model, which can be prohibitive.

**Representation learning for RL.** Reinforcement learning (RL) methods often learn representations as part of policy learning from reward signals. However, learning representations from solely reward signals can be highly sample inefficient, leading to poor generalization, and struggles in high-dimension state and action settings [41, 63, 74]. Consequently, techniques that learn improved representations have been shown to significantly improve RL performance [73, 19, 25, 40, 24]. Typically, these methods construct additional supervision via offline data or self-supervised objectives to drive representation learning [12, 11]. These methods have achieved significant success in a variety of settings, such as vision-based control and multi-task RL [10, 44, 30]. One such way of constructing supervision for representation learning is by leveraging future states and actions sampled from offline trajectories [48, 18, 60]. In particular, prior work has shown that increasing the similarity between state-action pairs sampled from the same trajectory can serve as a simple yet powerful way to learn improved representations for RL, that additionally acts as a type of goal-conditioning by making representations more similar along the same trajectory [20, 22]. Still, representation and goal-conditioned RL largely have not been considered for natural language domains.

## 3 Method

Drawing on recent efforts on goal-conditioned reinforcement learning [5, 13] and representation learning [20, 31] in other areas, we show that we can learn representations that approximate a Q-function within a language reward model using contrastive learning. Since the Q-function quantifies the expected cumulative future rewards of taking a specific action from a given state [47], forcing the reward model's output scores to be entangled with an approximated Q-function has the potential to improve the reward model's credit assignment and preference ranking accuracy. We first present preliminaries on reward modeling and contrastive learning in Section 3.1 and present our method in Section 3.2.

### 3.1 Preliminaries

**Preference ranking reward modeling for LMs.** Typically, a reward model is parameterized $r(x, y) \rightarrow \mathbb{R}$ to return a scalar reward, given prompt $x$ and completion sequence of tokens $y = [y_0, ..., y_T]$. Given a dataset $\mathcal{D}$ consisting of triples $(x, y^w, y^\ell)$ of preferred and dispreferred completions, $y^w$ and $y^\ell$, respectively, the reward model is optimized via a paired loss:

$$\mathcal{L}^R = -\frac{1}{|\mathcal{D}|} \mathbb{E}_{(x, y^w, y^\ell) \in \mathcal{D}} \log\left(\sigma(r(x, y^w) - r(x, y^\ell))\right). \tag{1}$$

Note, the reward model, $r(x, y)$, is trained to provide a scalar feedback on the *entire* completion $y$.

**Approximating Q-functions via contrastive learning.** In deep reinforcement learning, the Q-function quantifies the expected cumulative future rewards of taking an action from a given state and is parameterized by a neural network [47, 20]. Let $s_t$ be the current state from which a model can take an action $a_t$. Taking action $a$ at the current state and continuing on that trajectory will lead to some future state $s_k$. The future states reached along this trajectory can be positive, $s_k^+$, by reaching a target goal state or negative, $s_k^-$, by reaching a different or undesired state. A critic function $f$ can approximate a Q-function up to a multiplicative constant by optimizing a contrastive learning objective [20]:

$$\mathcal{L} = \log(\sigma(f(s_t, a_t, s_k^+))) + \log(1 - \sigma(f(s_t, a_t, s_k^-))). \tag{2}$$

Here, the critic function $f$ is a scoring function between a state-action encoder and state encoder.

### 3.2 Learning Goal-Conditioned Representations in Language Reward Models

In this section, we describe how to efficiently learn goal-conditioned representations using decoder-only transformer models [64, 54]. Additionally, we describe how to use these representations as Q-values that can be leveraged to detect generation errors and in guided decoding [33].

**Reward modeling through the bottleneck of a Q-function.** Given the prompt $x$ and a corresponding completion sequence $y$, we seek to learn a language reward model that both scores $y$ and has representations that can be used to approximate a goal-conditioned Q-function. This Q-function $Q_{y_g}^\pi(y_{\{0,...,t\}}, y_{t+1}), t \in [0, ..., T]$, captures the expected reward for a policy $\pi(y_{t+1}|y_{\{0,...,t\}}, x)$, for some goal state $y_g$ and next token $y_{t+1}$, where rewards are defined as likelihood of achieving the goal at the next token generation. Intuitively, the goal state could be an optimal generation from the policy that a human would find satisfactory, such as the correct solution to a math problem or a helpful response. The Q-function captures the expected rewards from choosing token $y_{t+1}$ at the current step, considering the prompt $x$ and tokens generated so far $y_{\{0,...,t\}}$.

We initialize our reward model, $r$, from an instruction-tuned causal LM. We consider this reward model to be decomposed into two components: 1) feature extractor $\phi$, which produces a representation for a given token; 2) a reward projection head $r'$ that predicts a reward score based on that token representation. We use $\phi$ to obtain representations of our state-action pair, $h_{t+1}$, and arbitrary future states, $h_k$:

$$h_{t+1} = \phi(y_{t+1}|y_{\{0,...,t\}}, x) \quad \text{and} \quad h_k = \phi(y_k|y_{\{0,...,k-1\}}, x). \tag{3}$$

Hence, we parameterize our reward model as

$$r(x, y) = r'(\phi(y_T|y_{\{0,...,T-1\}}, x)). \tag{4}$$

In practice, the feature extractor is the LM decoder and the representations that we use are the hidden token representations prior to predicting an output token. Meanwhile, the reward projection is either a linear layer or multi-layered perceptron (MLP) in our experiments.

With this reward model, we train the representations from the feature extractor $\phi$ using a contrastive loss [20], which becomes:

$$\mathcal{L}^C = \log(\sigma(f(y_{\{0,...,t\}}, y_{t+1}, y_g^+))) + \log\left(1 - \sigma(f(y_{\{0,...,t\}}, y_{t+1}, y_g^-))\right), \tag{5}$$

where $f(y_{\{0,...,t\}}, y_{t+1}, y_k)$ is the cosine similarity between the representations of the state-action pair $(y_{\{0,...,t\}}, y_{t+1})$ and a future state $y_k$. In Equation 5, $y_g^+$ and $y_g^-$ are our positive and negative

goal states, which we discuss how to obtain in practice later in this section. By forcing the reward scores predicted by $r'$ to depend on the representations from $\phi$, which is learned via this contrastive objective, the reward score incorporates a signal as to which actions are closer to a desired goal state.

We jointly train the reward model on the contrastive objective in Eq. 5 and the preference ranking objective optimized by Eq. 1. Thus, our final objective is given by

$$\mathcal{L} = \mathcal{L}^R + \lambda \cdot \mathcal{L}^C, \tag{6}$$

where $\lambda$ is a hyperparameter to balance the tradeoff between optimizing the contrastive loss and the rewards.

In the context of RLHF, given that we have preference labels between two or more generations for the same prompt and recalling that $\mathcal{L}^C$ approximates a Q-function, we can improve the term's alignment with human preferences by sampling positive goal states from preferred generations $y^w$ and negative goal states from dispreferred generations $y^\ell$.

**Computing contrastive loss in practice.** Given a preference-ranking instance from a training batch, let the preferred completion, $y^w$, and dispreferred completion, $y^\ell$, have lengths $T$ and $T'$, respectively. The representations of the positive state-action and goal state pair, $(y_s, y_g^+)$, are given by sampling two points, $i, j \sim [0, T], i < j$, from the preferred completion and retrieving hidden states $h_i^w$ and $h_j^w$. Similarly, the representations of the negative pair, $(y_{s'}, y_g^-)$, are given by sampling any index $m \sim [0, T]$ from the preferred completion and any index $n \sim [0, T']$ from the dispreferred completion and then retrieving hidden states $h_m^w$ and $h_n^\ell$. We note that source states $h_i^w$ and $h_m^w$ are sampled independently and are not necessarily equal. The cosine similarities between positive and negative pair representations are passed through the sigmoid function and averaged across the training batch. We will refer to this method as the Single Goal State (SGS) contrastive loss.

**Computing Q-values.** Once the reward model has been trained, we can use it to compute an additional *Q-value*, which gives us a measure proportional to the expected future rewards given the current state and a specific action. The Q-value at a token $t$ is the cosine similarity between the trained reward model's hidden state, $h_t$, and a goal state, $h_g$. The the choice of goal state here is flexible. For our experiments in Section 4.1, unless specified otherwise, we set the goal state to be the mean representation of all the preferred completion in the reward model training set. More explicity, the goal state is computed as:

$$h_g = \frac{1}{N} \sum_{y_w \in \mathcal{D}} h(y_w) \tag{7}$$

where $h(.)$ represents the reward model's last hidden layer of the final token of the completion. In Appendix C, we ablate this choice and find this to be the best method for computing the goal state during inference. Thus, we do not require any additional annotations aside from the starting preference ranking dataset. In experiments on LM steering, we construct a prototype to be used as the goal state. The prototype is constructed using generations labeled as helpful, complex, coherent, and correct by annotators in Nvidia's HelpSteer dataset [67]. Out of the highly-scored generations, we sample a subset and use the mean represetations of the sample as the goal state for the reward model. We ablate these choices in Appendix D.2.

## 4 Experiments

To explore our method, we experiment with two settings: First, mathematical reasoning with code (Section 4.1), where LMs must use reasoning and coding to answer mathematical questions. The ability to code has become increasingly important for LLMs as it enables a broad set of capabilities for reasoning (*e.g.,* tool use [58]). This line of experiments elucidates the ability of our method, and the resulting representations, to guide and improve step-by-step reasoning capabilities. Second, natural language alignment (Section 4.2), in which an LM's responses must align with certain desired properties, such as helpfulness and harmlessness. These experiments show the capacity of reward models trained with our method to steer LM outputs in a more open-ended setting. In particular, we focus on helpfulness, harmlessness, and other desirable characteristics (*e.g.,* correctness, coherence).

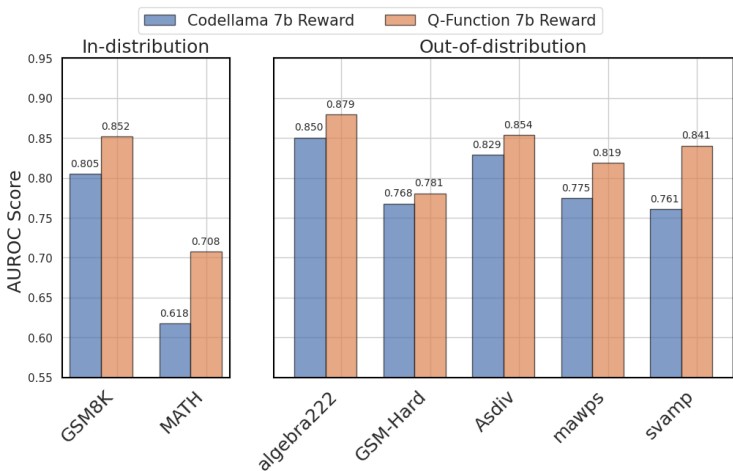

Figure 2: AUROC scores comparing the baseline Codellama 7b Reward vs. our proposed method Q-Function 7b Reward on the rewards attributed to the base-model greedy generations across several math benchmarks.

## 4.1 Mathematical Reasoning with Code

### 4.1.1 Experimental Setup

For training reward models, we use the OpenMathInstruct-1 dataset [62], a math instruction tuning dataset with 1.8M problem-solution pairs on the training sets of GSM8k and MATH, which are generated by the Mixtral-8x7B model [32]. To form our reward model training set, we create pairs of solutions for the same problem, where one solution is correct (*i.e.,* preferred) and the other is incorrect (*i.e.,* dispreferred).

Our base model is OpenMath-CodeLlama-7b-Python-hf [62], a strong decoder-only LM that has been instruction tuned on OpenMathInstruct-1 to use a Python interpreter in order to solve mathematical problems, which we add a reward head to predict reward scores (Section 3.2). This model trained with the traditional preference ranking objective serves as our baseline (**Codellama 7b Reward**), which we compare with the same model trained using our method (**Q-Function 7b Reward**).

We evaluate on several popular math benchmarks. Since the problems in the preference ranking dataset come from the training sets of GSM8k and MATH, we consider their respective test splits to be in-distribution (ID). We also evaluate on test sets we consider to be out-of-distribution (OOD), namely algebra222 [27], GSM-Hard [23], Asdiv [46], mawps [35], and svamp [52], to test for further generalization.

Detailed information on data, models, and training can be found in Appendix A.

### 4.1.2 Reward Modeling Evaluations

We obtain completions from the base model by greedy decoding, following prior work [62, 66], and measure the reward score AUROC. This quantifies the ability of a model's reward score to discern between correct/incorrect sequences across different thresholds. In Appendix E, we additionally assess Q-Function 7b Reward's ability to rank completions.

**Learned representations help reward scores discern correct/incorrect solutions.** In Figure 2, we see that the reward model trained with our representation learning approach consistently achieves higher AUROC scores compared to the model trained with only the standard preference ranking loss. On the ID datasets, we see a gain of approximately 0.05 and 0.09. Meanwhile, on the OOD datasets, the average improvement is 0.04, with maximum of roughly 0.08 on svamp. Although these datasets are math problems, they vary in the types and difficulties of questions asked. Despite being OOD, we see that our method improves the performance when judging these prompts and generations. Overall, these results suggest that the representations learned using our approach help improve the reward score predictions such that they are more accurate and generalizable.

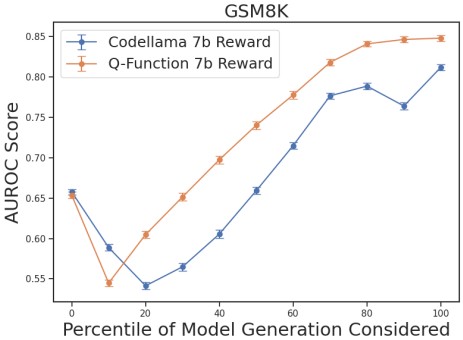
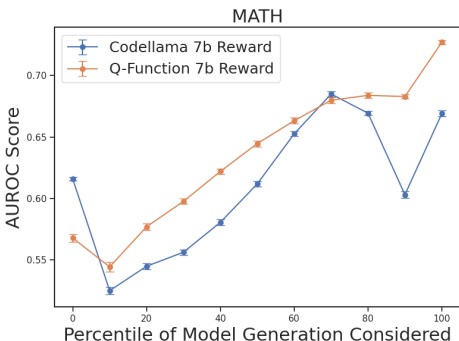

Figure 3: AUROC scores on the rewards attributed to partial base-model generations across 50 samples on GSM8k and MATH. The error bars depict the 95% confidence intervals (with sample size $n = 50$) at each percentile of generation considered. The Q-Function reward model has incremental increase in performance with more information, whereas, the traditional reward model's performance is a lot more varied in attributing intermediate rewards.

**These representations also help reward scores judge which partial solutions are promising.** Part of the motivation and design of our approach is to encode the likelihood of reaching a certain goal state at intermediate steps in the generated sequence. To examine how well our method accomplishes this, we measure reward score AUROC when the model is only shown up to a certain percentage of the generated sequence (*i.e.,* percentile). Here, we show this on the ID datasets, where we consider the 50 sample generations from the base model and plot the mean AUROC across the 50 samples. Plots for other benchmarks can be found in Appendix F. Looking at Figure 3, we find that the AUROC scores for both reward models tend to increase as they see more of the generated sequence, but Q-Function 7b Reward largely maintains higher AUROC across different percentiles and generally exhibits fewer significant drops between percentiles. These improvements across percentiles imply that the reward scores do indeed better capture which generated sequence are likely to be correct or incorrect.

### 4.1.3 Directly Using the Learned Representations

Our experiments thus far illustrate the effect of the learned representations on the reward scores. Here, we turn our attention to utilizing these representations directly to explore some of the encoded properties. In these experiments, we filter generations from the base model using the Q-values (*i.e.,* cosine similarities between intermediate and goal state representations). We take 50 base model generations for all the GSM8k and MATH test problems, which are provided by Toshiniwal et al., and we filter any generation that has a Q-value less than zero [62]. We compare this against a baseline of majority voting over the 50 generations.

**Q-values can prune incorrect sequences.** We find that filtering based on Q-values provides meaningful improvements in accuracy compared to majority voting over all 50 samples, based on Table 1. Specifically, the majority vote accuracy improves by 1.2% and 3.9% on GSM8k and MATH, respectively. Looking at the increased proportion of correct sequences after filtering, it appears that sequences with the negative Q-values generally correspond to incorrect sequences and are filtered out in this setting, thereby improving the accuracy. On the other hand, we also see that the average percentage of sequences that are filtered out can be quite large, such as 65.6% on the MATH dataset, and some correct solutions can also be filtered out. Nevertheless, using the Q-values based on these learned representations can help improve the accuracy of such majority voting schemes.

**Q-values can detect errors in reasoning.** Experiments on rewards scores for partial sequences (Section 4.1.2) suggest that the learned representations help capture the likelihood that a sequence will reach a goal state from an intermediate step. We can also qualitatively explore this property using the Q-values and the representations themselves. In Figure 4, we show examples that are filtered based on the Q-value filtering process above. In the left example, the Q-value drops when there is a mistake of multiplying the "`donuts_price_per_dozen`" by "`donuts_per_dozen`". The Q-value in the right example similarly shifts once the "`jazz_dance_percentage`" is naively added to the

|  | GSM8k | | | |
| --- | --- | --- | --- | --- |
|  | Accuracy (%) | Prop. Correct (%) | Avg. Filtered (%) | Avg. Tokens Saved (%) |
| Maj. @ 50 | 84.8 | 74.9 | - | - |
| Filtered | **86.0** | **84.0** | 35.9 | 25.5 |
|  | MATH | | | |
|  | Accuracy (%) | Prop. Correct (%) | Avg. Filtered (%) | Avg. Tokens Saved (%) |
| Maj. @ 50 | 55.6 | 40.2 | - | - |
| Filtered | **59.6** | **52.0** | 65.6 | 55.6 |

Table 1: Majority vote versus Q-value filtering performance. Accuracy shows the final majority voting accuracy on the respective benchmarks. Prop. Correct is the proportion of the correct class in the final sample set considered. Avg. Filtered is the average percentage (out of 50 total generations) that were discarded across all problems in the benchmark. Avg. Tokens Saved is the average percentage of tokens that are saved with the assumption that the remaining tokens after the first negative Q-value is discarded across all the problems in the benchmark. Both the difference in accuracy and proportion of correct class are t-test significant.

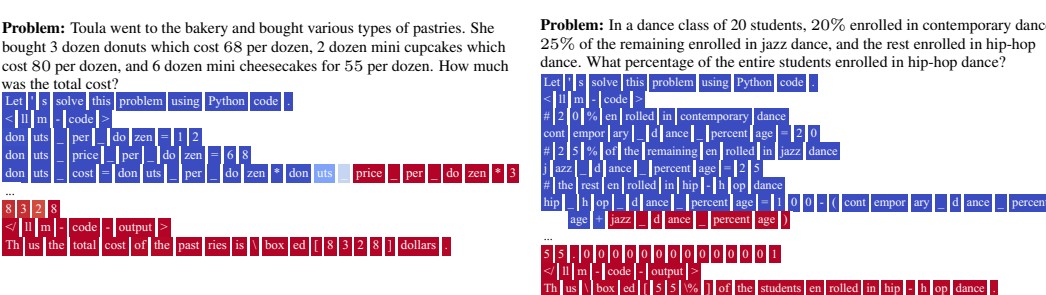

Figure 4: Two examples from the GSM8K dataset that was filtered via the Q-value pruning. The token level Q-values are portrayed as a heat map where the colors red and blue represents scores close to −1 and 1, respectively. Both examples illustrate that the Q-values pinpoint the exact logical error in reasoning. The full version of these examples can be found in Appendix G

"`contemporary_dance_percentage`". Overall, these examples illustrate the potential effectiveness of using the representations and resulting Q-values for localizing errors and identifying promising sequences.

### 4.1.4   Aligning Policy Models with RLHF

We investigate whether the reward model trained with our proposed method provides a better reward signal when performing RLHF, which is an important usage of these reward models. We compare the math problem solving performance of a policy model that is aligned using a preference-ranking reward model versus our Q-Function reward model in the Proximal Policy Optimization (PPO) [59] algorithm. We additionally compare the performance against DPO as it serves as a strong baseline [55]. The prompt dataset is given by problems in the training sets of GSM8k and MATH. To introduce diversity in the prompts, which has been shown to improve PPO, we also add problems from the MathQA dataset - a large scale dataset with real world math problems [4]. We keep the settings and hyperparameters during PPO constant, and further details are in Appendix A.

Table 2 shows that, given the same settings, utilizing a reward model trained with our method improves policy model accuracy across several benchmarks. We see an average gain, weighted by benchmark size, of 1.7%. Intuitively, this shows that the improvements we observe in reward scores (Section 4.1.2) do translate to improvements in the policy model via RL, which further supports the value of the representations learned via our approach. Although, the gains in accuracy are not as significant compared to those observed with the reward scores. A possible explanation for this is that, due to experimental constraints such as additional data collection, the preference-ranking dataset used for reward model training are off-policy Mixtral-8x7B model [32] generations. Hence, a clear next step for this setting would be to retrain the reward model with on-policy preference-rankings and iterate. Interestingly, we observe that DPO performs on par with our method for the ID tasks

(GSM8k and MATH). However, the policy trained with the contrastive RM performs much better on OOD tasks (algebra222, GSM-Hard, Asdiv, mawps, and svamp) when compared to DPO. These findings could be explained by related works that analyze DPO and find it can perform poorly in OOD settings [72]. All in all, these results indicate that PPO training with the contrastive RM leads to better generalization than training with DPO, particularly in OOD settings.

| Model | ID | | OOD | | | | |
|---|---|---|---|---|---|---|---|
| | GSM8k | MATH | algebra222 | GSM-Hard | Asdiv | mawps | svamp |
| Codellama Base | 75.9 | 43.6 | 65.6 | 60.1 | 77.7 | 93.5 | 79.6 |
| DPO | 80.1 | 44.8 | 64.8 | 59.9 | 76.9 | 90.3 | 76.6 |
| Codellama PPO | 79.3 | 43.4 | 65.8 | 61.1 | 77.4 | 91.6 | 78.5 |
| Q-Function PPO | **80.5** | **45.1** | **70.9** | **62.7** | **79.5** | **93.6** | **81.2** |

Table 2: Accuracy of the policy model trained via PPO using a preference ranking reward model vs our Q-Function reward model. We present average accuracy across 4 independent runs. The base model results are also shown as a reference point presented in [62] and DPO results are shown as a strong baseline. The full table with confidence intervals is in the appendix (Table 6).

## 4.2 Natural Language Alignment

While fine-grained feedback (and subsequently, Q-values) are most sensible in reasoning environments with explicitly incorrect logical steps, a natural extension of the experiment is to apply our approach to reward models in the natural language setting, where we expect models to learn improved representations with respect to their alignment towards human preferences. We first explore this for training a reward model for helpfulness and harmlessness [8]. Then, we explore if the learned representations can be further used for steering language model at inference time, via guided decoding, to generate sequences that are more helpful, complex, or coherent. The backbone of our reward model is Llama 3-8B-Instruct [1], and detailed information can be found in Appendix B.2.

### 4.2.1 Improving Reward Modeling for Helpfulness and Harmlessness

We use the Helpful-Harmless dataset [8] for training and evaluation. We select this dataset because of it's large scale, established representation in the RLHF community, and general purpose nature which is helpful to avoid restricting our natural language reward model towards a specific domain. Similar to our previous experiments, we compare training with our approach versus standard reward model training. Table 3 (left) shows that training with our method also improves reward scores on this natural language data, increasing the ability of the reward model to discern between helpful and unhelpful responses.

### 4.2.2 Reward Model Representations can Steer Language Models Towards Helpful Generations

A practical extension of learning stronger natural language representations is using the extensions for steering the LM. To test this, we experiment with guided decoding [33, 70] with Llama-3-8B-Instruct, which defines a decoding objective function to guide inference during beam search. In traditional guided decoding (and our SFT baseline), the confidence score of the model is extracted from an additional inference step on each beam. In our method (Q-Function Prototype), the confidence score is the cosine similarity between the sequence embedding from the reward model and the hidden state [20, 2]. We construct the prototype by embedding 20 examples that score highly in all categories (helpfulness, correctness, coherence, complexity) using HelpSteer, a model alignment and steering dataset [67]. The reward model is trained using HelpfulHarmless as per previous sections [8]. Details for guided decoding are in the Appendix B.

Table 3 (right) shows our results, and evaluation is done via LLM-as-a-judge (GPT-4) [75]. Using the representations fine-tuned by the contrastive loss improves across all categories and generally steers generations towards more helpful generations. We find that complexity improves the most, suggesting that the generations produced by the using the prototype helps the model avoid beams that are too simple. Suprisingly, the correctness of the responses is increased also, suggesting that

although the SFT prototype usually prefers beams that are simple and correct, the prototype succeeds at picking beams with added complexity without sacrificing correctness or helpfulness.

| Model | HH |
| --- | --- |
| Llama 8b Reward | 68.2 |
| Q-Function 8b Reward | **70.5** |

| Model | Helpful | Correct. | Coherent | Complex. |
| --- | --- | --- | --- | --- |
| SFT | 45.2 | 43.9 | 41.8 | 39.2 |
| Prototype | **54.8** | **56.1** | **58.3** | **60.8** |

Table 3: Comparison of model performances in different settings. (a) Reward model accuracy on the Helpful-Harmless test set. Our result highlights that the representations learned by the Q-function are helpful for discerning between desirable and undesirable sequences in the natural language setting. (b) Winrate (%) vs. SFT model. In the Q-Function prototype setting, we use the cosine similarity of the sequence and the prototype (made from 20 examples that score high on all 4 categories) to score each beam. In the SFT setting, the model is fine-tuned on the same examples as the prototype and guided decoding is run via model scores. Evaluation is done via GPT-4.

## 5 Limitations and Future Work

**Deriving more informative goal states.** In Appendix D.2, we demonstrate the importance of having a precise and meaningful goal state. At train time, our SGS method achieves this by looking at future tokens of the preferred and dispreferred completions. During inference, however, we take the mean representation of multiple goal state completions from the training data. Although we see that the SGS method generalizes despite this discrepancy, future work could investigate other methods to derive more precise inference-time goal states.

**Partial Q-values correlate with partial reward scores.** An interesting observation, shown in Appendix H, is that the reward scores for *partial* completions are correlated with the Q-values. While our experiments show that this can benefit reward scores, especially for math reasoning, disentangling these values may be desirable in some instances. For example, one may want to incorporate the completeness of the generated sequence in the reward score, such as for creative writing, meaning incomplete sequences should strictly get low reward scores. However, currently, partial sequences may still score highly if they are likely to reach a goal state. This is expected as the final reward score is derived by a linear projection from the hidden representation, while the Q-values are derived as the cosine similarity between that same representation and goal state. In Appendix H, we explore the effects of learning an MLP projection for the rewards to help decouple the two signals. Further disentangling these signals in the representations could be an interesting direction for future work.

**Further advancing policy models.** A clear line of future work here is to integrate the additional Q-value signal that we gain from this method into the RL algorithm to further improve policy model training. Future work should also investigate the impacts of iteratively retraining the reward model on on-policy completions and the policy model on the updated reward model.

## 6 Conclusion

Learning proper representations in the reward model is essential for the model's generalization and ability to produce accurate rewards during RLHF. We introduce a method of applying goal-conditioned Q-functions to learn representations that capture the expected reward for a goal-conditioned policy. On math benchmarks, such as GSM8k, we improve performance and show that the reward model has a greater ability to discern between correct and incorrect solutions. On natural language alignment experiments, we show improvement in reward model performance and that the embedding can be used for natural language steering. Our findings show that using Q-values during reward model training can improve the representations of the reward model and suggest promising future directions for further advancing language models via RLHF.

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

# Appendix to
# Learning Goal-Conditioned Representations for Language Reward Models

## A  Mathematical Reasoning with Code Details

In this appendix, we discuss further experimental details for our results in Section 4.1.

### A.1  Reward Model Training and Evaluation Details

**Preference ranking dataset.**  To construct the preference ranking dataset, we used the OpenMathInstruct-1, a math instruction tuning dataset with 1.8M problem-solution pairs generated by the Mixtral-8x7B model. The questions come from a blend of the training subsets for GSM8k and MATH. We matched up generations that had the final answer output correct vs. incorrect as the preferred and dis-preferred generation. Some problems are used several times, but all preferred and all dis-preferred solutions are unique. In total, we curated $257,584$ total preference rankings over a total of $12,518$ unique problems.

**Prompt format.** To prompt the base model to answer a math question, we use the same Nemo prompt format that was used to fine-tune the model. This prompt format was used both for reward model training, PPO training, and all evaluations. The following is the prompt format:

> System:
>
> You're an expert Python programmer and mathematician. Help the user to solve this problem using code when necessary. Make sure to put the answer (and only answer) inside \boxed[].
>
> User: **{problem}**
>
> Assistant:

where **{problem}** is replaced with a math problem from the dataset.

**OpenRLHF library.** We implement the reward model training with contrastive loss and the baseline of standard preference ranking using the OpenRLHF library [29] in combination with PyTorch [51]. We made the following modification to the OpenRLHF Library:

- A new contrastive loss method that computes the contrastive objective
- Optionally add auxiliary loss to the reward model training objective if experimenting with our reward model method
- Optionally change the reward model reward head from a linear projection to a multi-layer perceptron. The MLP layer is a simple 3 fully connect layer with ReLU activation in between, all with the same hidden size as the representation dimension.

**Reward model training parameters.** We train both the baseline reward model and the Q-Function reward model as well as any reward model ablations for 1 epoch with batch size of 64.

The hyper-parameter for the contrastive lambda of $0.5$ was chosen by doing a sweep of $0.1, 0.5, 0.9$, and found that the $0.5$ setting had the most stable to training loss as well as training rewards for preferred vs. unpreferred completions.

**Reward model evaluation.** We evaluate the Reward Model performance on the test splits of several math benchmarks: GSM8K [16], MATH [4], algebra222 [27], GSM-Hard [23], Asdiv [46], mawps [35], and svamp [52]. To get a set of completions on each problem in these set of benchmarks, we use the greedy generation of the base model along with annotations of whether each completion provides the correct answer, which are provided by Toshiniwal et al. [62]. Then for each completion, we first format using the Nemo format template described earlier, and use the Reward Model to predict a reward score. In predicting the reward score, we also retrieve the last hidden state of the Reward Model as well as the predicted reward score for each token in the completion, not including

| hyperparameter and settings | value |
| --- | --- |
| batch size | 64 |
| learning rate | $9 \times 10^{-6}$ |
| representation dimension | 4096 |
| contrastive lambda ($\lambda$) | 0.5 |
| num epoch | 1 |
| preference ranking loss function | sigmoid |
| optimizer | Adam |
| seed | 42 |

Table 4: Hyper-parameters for Reward Model training

the prompt tokens. Utilizing the Reward Model predicted reward score and the annotation of the correctness of each completion, we compute an AUROC score using the Python scikit-learn package [53]. Using the same procedure, we plot the partial AUROC scores at every tenth percentile of each completion. The AUROC scores are compared between Q-Function 7b Reward (our method) and Codellama 7b Reward and the results are summarized in Section 4.1.2.

## A.2 Learned Representations Experiment Details

**Computing Q-values.** In order to retrieve the Q-values, we take the cosine similarity between the goal state and the hidden states for each token, which is retrieved during the forward pass when collecting the reward scores. The goal state in all the experiments is the mean representation of all the preferred completion in the preference ranking dataset. More explicitly, we pass each preferred problem and response pair through the trained Reward Model and then we take the average across the last hidden state of the final token of each problem and response pair.

**Filtering sampled generations.** In order to understand the value of these learned representations, we use these Q-values to filter generations. For each of the In-Distribution datasets, GSM8K [16] and MATH [4], we use the 50 completion samples provided by Toshiniwal et al. [62]. We take each completion and we discard it if the Q-value for any of the tokens is less than 0, which indicates that the expected future rewards from that state is low. This is performed for each problem and each generation sample. Using the final set of remaining completions, we then compute the majority vote accuracy, comparing it against the baseline of majority vote at 50. Furthermore, in order to visualize these Q-values, we plot a heat map of these values on the corresponding tokens. The results and plots from this experiment is detailed in Section 4.1.3.

## A.3 RLHF Training and Evaluation Details

In these set of experiments, we investigate whether the reward model trained with our proposed method provides a better reward signal when performing RLHF. Here we explain further experimental details for the results in Section 4.1.4.

**Prompt dataset.** The prompt dataset for PPO comprises of the training problems from GSM8k and MATH, which constitutes a total of 12,518 problems [16, 28]. We also incorporate all the problems from the MathQA dataset [4], which has 37,297 unique problems. Thus, in total, the prompt dataset for PPO has 49,815 unique problems. In addition, at each training step of the policy model, we add a small pretraining coefficient. The SFT dataset for the pre-training coefficient is comprised of all the preferred completions in the preference ranking dataset.

**PPO training parameters.** We also use the OpenRLHF package and PyTorch for PPO training [29, 51]. No major code changes were made to the OpenRLHF package, with the exception of the dataset loader for our custom prompt dataset. Table 5 summarizes the hyperparameters used for all PPO training reported in this paper.

**Policy model evaluation.** We evaluate the policy-trained model on the test set on all the aforementioned math benchmarks. For each problem in these benchmarks, we process the input to use the prompt format detailed above and perform zero shot evaluation. We perform 4 independent runs to account for variance. In order to perform this evaluation, we utilize the Nemo-Skill [62] package. As

| hyperparameter and settings | value |
|---|---|
| train batch size | 128 |
| number of episodes | 1 |
| rollout batch size | 512 |
| training epochs | 1 |
| actor learning rate | $3 \times 10^{-6}$ |
| critic learning rate | $9 \times 10^{-6}$ |
| init kl coeff | 0.02 |
| normalize reward | true |
| generation max length | 1024 |
| GAE gamma | 1.0 |
| GAE lambda | 0.95 |
| policy model temperature | 1 |
| pretraining coefficient | 0.05 |
| learning rate | $9 \times 10^{-6}$ |
| representation dimension | 4096 |
| contrastive lambda ($\lambda$) | 0.5 |
| optimizer | Adam |

Table 5: Hyper-parameters for PPO training

a pre-requesite to performing evaluation on this package, we first had to convert the model to the Nemo model format. Table 6 displays the average accuracy scores across 4 independent runs, as well as the 95% confidence interval. The variability across the 4 runs comes from random seeding, which influences the initialization of the Value Model as well as dataset shuffling.

| Reward Model | IID | | OOD | | | | |
|---|---|---|---|---|---|---|---|
| | GSM8K | MATH | algebra222 | GSM-Hard | Asdiv | mawps | svamp |
| DPO | $80.1 \pm 0.1$ | $44.8 \pm 0.2$ | $64.8 \pm 0.8$ | $59.9 \pm 0.4$ | $76.9 \pm 0.0$ | $90.3 \pm 0.2$ | $76.6 \pm 0.1$ |
| Codellama | $79.3 \pm 0.2$ | $43.4 \pm 0.2$ | $65.8 \pm 1.6$ | $61.1 \pm 0.3$ | $77.4 \pm 0.2$ | $91.6 \pm 0.3$ | $78.5 \pm 0.9$ |
| Q-Function | $\mathbf{80.5 \pm 0.3}$ | $\mathbf{45.1 \pm 0.1}$ | $\mathbf{70.9 \pm 1.7}$ | $\mathbf{62.7 \pm 0.5}$ | $\mathbf{79.5 \pm 0.5}$ | $\mathbf{93.6 \pm 0.3}$ | $\mathbf{81.2 \pm 0.4}$ |

Table 6: Average accuracy and 95% confidene interval after the base policy model was trained via DPO, PPO using a preference ranking reward model, and PPO using our Q-Function reward model, across 4 different independent runs.

**Code execution setup.** Our sandbox environment is constructed using Python 3.10 and includes necessary mathematical libraries such as NumPy to support the execution of LM-generated code [26]. This environment is containerized and deployed across a Kubernetes cluster with 64 replicas, designed to handle the training and evaluation workload in parallel. Each replica is allocated 1 CPU and 1 Gi of memory to efficiently manage the computational demands. Each sandbox establishes communication and accepts requests through a Flask server.

**Code execution security considerations.** In our current evaluation setup, the code generated by the Language Model is primarily focused on solving mathematical questions and originates from trusted sources, allowing us to maintain a controlled and relatively secure environment. However, extending this methodology to broader applications, where code is executed from varied or unknown sources, significantly increases security concers such as malware intrusion and unauthorized data access. To mitigate these risks, it is imperative to create a sandbox environment to isolate the code execution from the regular operational environment. Key protocols include enforcing least privilege in the sandbox to limit permissions, minimizing external communications to reduce attack surfaces, and rigorously validating both incoming code and outgoing artifacts to ensure they adhere to security best practices.

## A.4 Compute and Statistical Significance

For the reward model training and PPO, we used a single node, 8 gpus, and 88 CPU cores, 80 Gi GPU memory and 1TB system memory. The total reward model training time ranged between 6-7 hours

and the total PPO training was approximately 13-14 hours. We additionally used flash attention 2 and deepseed with zero_stage 3, for reward model training, and zero_stage 2 for PPO training [17, 3].

Moreover, all the experiment results presented are statistically significant, performed by the scipy library [65]. Where applicable, all error bars, whether a table or plot, are 95% confidence intervals, calculated via numpy package [26].

# B Natural Language Alignment Details

In this appendix, we discuss further experimental details for our results in the Section 4.2.

## B.1 Training a Natural Language Reward Model

**Dataset.** To train our natural language reward model baseline and contrastive reward model, we use the training set of Helpful-Harmless [8]. Helpful-Harmless is a popular natural language alignment dataset created by Anthropic with human-assistant dialog chains. Each pair contains a chosen and rejected response, where the chosen response generally aligns on axes of helpfulness and harmlessness. This dataset contains 169,352 rows where each row contains a pair of chosen and rejected samples.

**Training.** For training the reward model, we use the OpenRLHF [29] package and Pytorch. No major code changes were made to the OpenRLHF package, and we use the dataset loader given by OpenRLHF to load the Helpful-Harmless dataset for training. Table 7 shows the hyperparameters used during training for reproducibility.

| hyperparameter and settings | value |
|---|---|
| batch size | 64 |
| learning rate | $1 \times 10^{-4}$ |
| representation dimension | 4096 |
| contrastive lambda ($\lambda$) | 0.5 |
| num epoch | 1 |
| preference ranking loss function | sigmoid |
| optimizer | Adam |
| seed | 0 |

Table 7: Hyper-parameters for Natural Language Reward Model training

**Evaluation.** To evaluate the natural language reward model, we use the evaluation script in Open-RLHF [29]. We evaluate on the test set of Helpful-Harmless. For each chosen, rejected pair, we count the reward model score as correct if the chosen generation has a higher score than the reject pair.

## B.2 Guided Decoding with Contrastive Reward Model

**Dataset.** For guided decoding, we consider several different datasets. We considered OpenAssistant [36], a large corpus of assistant dialog and Prism-Alignment [34], a conversation alignment dataset. We also considered HelpSteer [67], a dataset created by Nvidia with labels on how helpful, coherent, correct, verbose, and complex a given response is. We settled on HelpSteer as our dataset for experimentation due to the relatively high-quality of the dataset (human-collected vs automatic) and the fine-grained category labels. The dataset contains 37,120 examples in the training set.

**Guided decoding.** We do not further train the model, but rather randomly sample 20 examples from each of the fine-grained category labels in HelpSteer: coherent, complexity, correctness, and helpfulness. We do not use the verbosity category because the hidden states are averaged at inference time, thus, there is a worry that we would lose the information needed to identify a sequence as verbose. We use SelfEval Guided Decoding [69] for running the decoding experiments. Equation 8 shows the objective function $\varepsilon(s^{1:T})$ used in guided decoding beam search.

$$\varepsilon(s^{1:T}) = \prod_T P_{LM}(s^t|s^{1:t-1})\rho C(s^t) \tag{8}$$

$P_{LM}$ is the language model distribution, $C(s^t)$ is the confidence score, and $\rho$ is a hyperparameter.

For the steering experiments, we set the max number of tokens per beam to 52, the beam search temperature to 0, temperature as 0.2, the number of samples to 2, and the confidence ratio to 0 e.g. equal weight to the confidence score and LM loss.

**Evaluation.** We evaluate on the test set of HelpSteer, which contains 1790 examples. For evaluation, we use GPT-4, asking GPT-4 to compare the SFT model and the model with the decoding prototype which generation is more helpful, correct, coherent, and complex. We ask GPT-4 to give a single token (0 or 1). For statistical significance, we run GPT-4 5 times for each evaluation example and take the majority. In addition, we find that for a small percent of cases GPT-4 will not product a single token even when asked. In this case, we throw away the example that did not produce a properly formatted response when computing averages.

### B.3 Compute and Statistical Significance

Natural language alignment training jobs were done on 1 core, 8 GPUs, 88 CPU cores, 1TB system memory, and 80GB GPU memory. An average training time for 1 episode of each prompt was around 13-14 hours.

All experiments are statistically significant. For the Helpful-Harmless experiments, the experiments are evaluated on a test set of 8.55k rows. For HelpSteer, the validation set is 1.79k rows.

## C  Inference-Time Goal State Ablation

In this appendix section, we explore different choices of inference-time goal states and compare their performance. To ablate this, we conduct a new experiment under the mathematical reasoning with code setting (Section 4.1) where we evaluate the effect of (a) choosing poor examples to construct the goal state and (b) the number of examples used to construct the goal state. In particular, we evaluate 3 settings for picking the sample completions for constructing the goal state:

1. We vary the number of preferred completions used to construct the goal state to evaluate the effect of adding more examples of good reasoning.

2. We vary the number of dispreferred completions used to construct the goal state, to evaluate whether bad examples lead to poor performance.

3. We incrementally add more dispreferred completions to a fixed sample of preferred completion in order to measure the robustness of our goal state computation to negative examples. We refer to the addition of dispreferred completions on top of all preferred completions as adding "corrupted" examples.

The results are in Figure 5. Overall, these results show that negative or unhelpful completions degrade performance, indicating the importance of choosing relevant examples for computing the goal state. Additionally, they demonstrate that having more examples of the concept leads to better performance as more generations are filtered with comparable proportion of those remaining generations being correct.

We additionally ablate our choice of using the last token in the completion sequence to construct the goal state. In this experiment, we repeat our filtering experiments from Section 4.1.3. Except, we randomly sample a token from in the completion sequence and use it as the goal state:

| Sampling Method | GSM8k | | MATH | |
|---|---|---|---|---|
| | Accuracy (%) | Prop. Correct (%) | Accuracy (%) | Prop. Correct (%) |
| Last Token | 86.0 | 84.0 | 59.6 | 52.0 |
| Random Token | 85.9 | 81.8 | 57.3 | 45.3 |

Table 8: Majority vote accuracy and average proportion correct when filtering examples using last token and random token sampling for constructing the goal state.

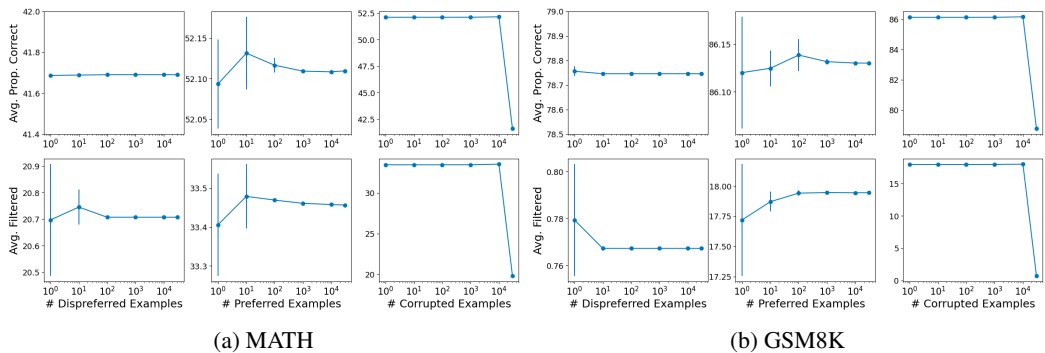

(a) MATH          (b) GSM8K

Figure 5: We plot the average proportion of non-filtered responses that are correct and the average number of completions (out of a total 50) that are filtered, as we vary the sample size (1, 10, 100, 1K, 10K, and 30K) of dis-preferred, preferred, and corrupted examples when constructing the goal state. All of the examples are retrieved from the preference ranking training dataset. We plot the performance metrics for the MATH and GSM8K test sets. Each of the test sets has 50 completions per problem that is generated by the base model and we filter any completion for which the Q-value for any token in the completion sequence is below 0. At each sample size, we run 3 independent random sample from the full set of preferred and dis-preferred completions to construct 3 independent goal states, with 95% CI.

From the result of this ablation, we see that last token sampling has superior performance, likely because the last token on average carries the most information about what good & complete generations look like.

# D   Train-Time Goal State Ablation

## D.1   Exploring Sources of Goal States

A natural question about our approach is: *What is the impact on the choice of goal state representation?* For sampling negative goal states, we experiment with randomly sampling any arbitrary completion (not necessasarily the same prompt) vs. the corresponding dispreferred completion to show the effect of picking completions that have similar contexts and invalid reasoning steps when constructing the contrastive loss. Furthermore, we experiment with taking the average of the hidden states of the positive and negative goal states across the train batch and then apply cosine similarity with the current token representation. The results of these experiments are in Table 9.

| Method | IID | | OOD | | | | |
|---|---|---|---|---|---|---|---|
| | GSM8k | MATH | algebra222 | GSM-Hard | Asdiv | mawps | svamp |
| Pref. Rank | 0.805 | 0.618 | 0.850 | 0.768 | 0.829 | 0.775 | 0.761 |
| Q-Function (SGS) | **0.852** | **0.708** | **0.879** | **0.781** | **0.854** | **0.819** | 0.841 |
| Q-Function (RS) | 0.801 | 0.626 | 0.832 | 0.716 | 0.723 | 0.760 | 0.802 |
| Q-Function (AVG) | 0.835 | 0.707 | 0.851 | 0.765 | 0.839 | 0.773 | **0.863** |

Table 9: Comparison of AUROC scores for different training time contrastive goal states on several popular math benchmarks. Pref. rank is the reward model trained with the standard preference-ranking objective, Q-Function (SGS) is trained with the method described in Section 3.2, Q-Function (RS) trained with the randomly sampled goal-state and Q-Function (AVG) trained with the batch averaged goal state.

The ablation in Table 9 shows that the positive and negative goal state representations we use are largely more effective than randomly sampled and batch averaged goal representations. Somewhat similar observations have been made for other contrastive learning approaches where negatives, especially hard-negatives, can play an important role [14, 57]. We see that when we randomly sample positive and negative goal states, there is a clear regression in performance on the OOD datasets,

even below the standard preference ranking trained reward model. Meanwhile, the batch averaged goal states offers competitive but largely lower performance compared to our final setup. Given these results, we use the SGS contrastive loss method as our final setup.

## D.2 Source State and Goal State Sampling Ablation

In this section, we ablate the sampling strategy for the source and goal states. In computing the contrastive loss during training, we need to sample a positive source and goal state pair as well as a negative source and goal state pair. To investigate how we should sample these tokens, we explore 3 different approaches.

**Method 1: Random Source and Goal State.** In this approach, the positive and negative source states and sampled randomly from the preferred response, then the positive goal state is sampled randomly amongst any token *after* the positive source state, and last the negative goal state is randomly sampled anywhere from the dis-preferred response. This is the method we describe in Section 3.2.

**Method 2: Late Goal State.** In this approach, we sample the positive and negative goals states only towards the end of the preferred and dis-preferred response, respectively. The intuition here is that the token later in the sequence encode more information about the whole response and thus provide more information as a goal state. In practice, we restrict the goal state sampling to come after the 90th percentile with respect to the total number of tokens in the response.

**Method 3: Late Source and Goal State.** In this approach, we sample both the source state and goal state towards the end of the preferred and dis-preferred response (after 90th percentile of tokens generated).

| Sampling Method | IID | | OOD | | | | |
|---|---|---|---|---|---|---|---|
| | GSM8k | MATH | algebra222 | GSM-Hard | Asdiv | mawps | svamp |
| Random Source and Goal State | **0.852** | **0.708** | **0.879** | **0.781** | 0.854 | 0.819 | **0.841** |
| Late Goal State | 0.841 | 0.696 | 0.821 | 0.775 | **0.860** | 0.729 | 0.761 |
| Late Source and Goal State | 0.829 | 0.643 | 0.867 | 0.779 | 0.829 | **0.831** | 0.833 |

Table 10: Comparison of AUROC scores for different source and goal state sampling method.

The ablation in Table 10 shows that the Random Source and Goal State sampling on average performs the best across both in-distribution and out-of-distribution tasks. We also see that with Late Goal State sampling, we have similar performance to Random Source and Goal State sampling for the in-distribution tasks but we notice a performance drop across several out-of-distribution benchmarks. The opposite is true for the Late Source and Goal State sampling where performance on the in-distribution benchmark drop and the performance out-of-distribution benchmarks are comparable to the random sampling. Given these results, we use the Random Source and Goal State sampling method as our final setup, which is equivalent to the (SGS) method referred to in the rest of the paper.

# E    Reward Model Ranking Performance

In this appendix section, we compare the Q-Function 7b Reward to Codellama 7b Reward by judging their respective abilities to rank completions. Since we have a total of 50 generations per problem, we use the baseline reward mode and our reward model to select the Top 1, 5, 10, and 25 samples as ranked by the reward scores. With the selected sample, we perform majority vote and also note the average proportion of the sample K that are correct solutions. The results are provided in Table 11. We show the Majority Vote Accuracy and Average Proportion Correct when selecting Top-K completions in order of reward model scores. The two reward models used here are the baseline Codellama RM and our Q-Function RM. The results show that our Q-Function RM clearly outperforms the baseline.

| Model | Top-K | GSM8k | | MATH | |
|---|---|---|---|---|---|
| | | Accuracy (%) | Prop. Correct (%) | Accuracy (%) | Prop. Correct (%) |
| | 1 | 84.6 | 84.6 | 51.7 | 51.7 |
| Q-Function RM | 5 | 86.2 | 84.5 | 59.7 | 51.1 |
| | 10 | 86.2 | 83.8 | 59.5 | 50.3 |
| | 25 | 85.5 | 81.8 | 57.8 | 47.0 |
| | 1 | 80.8 | 80.8 | 43.8 | 43.8 |
| Codellama RM | 5 | 84.3 | 81.9 | 54.0 | 45.8 |
| | 10 | 85.2 | 82.1 | 56.0 | 46.9 |
| | 25 | 85.8 | 81.1 | 56.5 | 46.1 |

Table 11: Majority Vote Accuracy and Average Proportion Correct when selecting Top-K completions in order of reward model scores. The two reward model use here is the baseline Codellama RM and our Q-Function RM.

## F  Partial Completion Rewards

In this appendix, we show the plots for the remaining out-of-distribution datasets, namely, algebra222, Asdiv, GSM-Hard, mawps, and svamp. The general trend observed in Section 4.1.2 still holds, where, Q-Function 7b Reward model incrementally improves as percentile of model generation considered increases. On the other hand, there is a lot more variability in Codellama 7b Reward partial auroc scores. More importantly, the performance when considering partial solutions is generally more accurate for the Q-Function 7b Reward. This demonstrates that the improved judgement of partial solutions that we observed for the in-distribution tasks extend to out-of-distributions tasks as well.

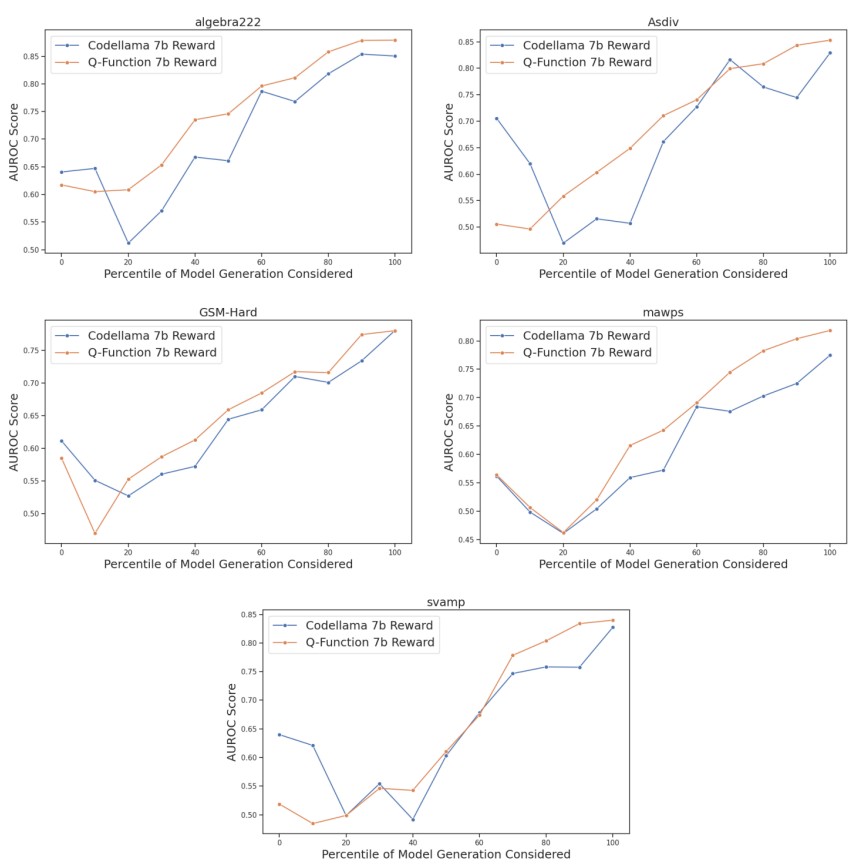

Figure 6: AUROC scores on the rewards attributed to partial base-model greedy generations on the OOD datasets.

# G    Q-Value Visualization

In this appendix, we display the full Q-value heat map for the examples shown in Figure 4. We also show additional examples that show the Q-values correctly scoring reasoning errors as well as some examples where the Q-values are noisy, unable to explicitly pin-point the logical error.

## G.1    Good Examples

**Problem:** Toula went to the bakery and bought various types of pastries. She bought 3 dozen donuts which cost 68 per dozen, 2 dozen mini cupcakes which cost 80 per dozen, and 6 dozen mini cheesecakes for 55 per dozen. How much was the total cost?

```
Let's solve this problem using Python code.
<llm-code>
donuts_per_dozen = 12
donuts_price_per_dozen = 68
donuts_cost = donuts_per_dozen * donuts_price_per_dozen * 3
cupcakes_per_dozen = 12
cupcakes_price_per_dozen = 80
cupcakes_cost = cupcakes_per_dozen * cupcakes_price_per_dozen * 2
cheesecakes_per_dozen = 12
cheesecakes_price_per_dozen = 55
cheesecakes_cost = cheesecakes_per_dozen * cheesecakes_price_per_dozen * 6
total_cost = donuts_cost + cupcakes_cost + cheesecakes_cost
total_cost
</llm-code>
<llm-code-output>
8328
</llm-code-output>
Thus the total cost of the pastries is \boxed{8328} dollars.
```

**Problem:** Marissa is hiking a 12-mile trail. She took 1 hour to walk the first 4 miles, then another hour to walk the next two miles. If she wants her average speed to be 4 miles per hour, what speed (in miles per hour) does she need to walk the remaining distance?

```
Let's solve this problem using Python code.
<llm-code>
distance_traveled = 4 + 2
time_taken = 1 + 1
average_speed = 4
remaining_distance = 12 - distance_traveled
time_remaining = remaining_distance / average_speed
time_remaining
</llm-code>
<llm-code-output>
1.5
</llm-code-output>
Thus Marissa needs to walk at \boxed{8} miles per hour to achieve her average speed of 4 miles per hour.
```

**Problem:** In a dance class of 20 students, 20% enrolled in contemporary dance, 25% of the remaining enrolled in jazz dance, and the rest enrolled in hip-hop dance. What percentage of the entire students enrolled in hip-hop dance?

```python
Let's solve this problem using Python code.
<llm-code>
# 20% enrolled in contemporary dance
contemporary_dance_percentage = 20
# 25% of the remaining enrolled in jazz dance
jazz_dance_percentage = 25
# the rest enrolled in hip-hop dance
hip_hop_dance_percentage = 100 - (contemporary_dance_percentage + jazz_dance_percentage)
# total number of students
total_students = 20
# number of students enrolled in hip-hop dance
hip_hop_dance_students = total_students * hip_hop_dance_percentage / 100
# percentage of students enrolled in hip-hop dance
hip_hop_dance_percentage_of_total = hip_hop_dance_students / total_students * 100
hip_hop_dance_percentage_of_total
</llm-code>
<llm-code-output>
55.00000000000001
</llm-code-output>
Thus \boxed{55\%} of the students enrolled in hip-hop dance.
```

**Problem:** When Freda cooks canned tomatoes into sauce, they lose half their volume. Each 16 ounce can of tomatoes that she uses contains three tomatoes. Freda's last batch of tomato sauce made 32 ounces of sauce. How many tomatoes did Freda use?

```python
Let's solve this problem using Python code.
<llm-code>
# each can of tomatoes has 3 tomatoes
tomatoes_per_can = 3
# each can of tomatoes has 16 ounces
ounces_per_can = 16
# Freda's last batch of sauce made 32 ounces
sauce_made = 32
# Freda used M cans of tomatoes
M = sauce_made / ounces_per_can
# Freda used N tomatoes
N = M * tomatoes_per_can
N
</llm-code>
<llm-code-output>
6.0
</llm-code-output>
Thus Freda used \boxed{6} tomatoes.
```

### G.2 Noisy Examples

**Problem:** Lee used to be able to run the 400-meter hurdles two seconds faster than Gerald would run the 400-meter hurdles. But Gerald changed his diet, which improved his speed by 10%. If Lee runs the 400-meter hurdles in 38 seconds, how fast can Gerald, with his improved diet, run the 400-meter hurdles, in seconds?

```
Let's solve this problem using Python code.
<llm-code>
lee_time = 38
lee_speed = 400 / lee_time
gerald_speed = lee_speed - 2
gerald_speed_with_improvement = gerald_speed * 1.1
gerald_time = 400 / gerald_speed_with_improvement
gerald_time
</llm-code>
<llm-code-output>
42.648709315375974
</llm-code-output>
Thus Gerald can run the 400-meter hurdles in \boxed{43} seconds.
```

**Problem:** John drives for 3 hours at a speed of 60 mph and then turns around because he realizes he forgot something very important at home. He tries to get home in 4 hours but spends the first 2 hours in standstill traffic. He spends the next half-hour driving at a speed of 30mph, before being able to drive the remaining time of the 4 hours going at 80 mph. How far is he from home at the end of those 4 hours?

```
Let's solve this problem using Python code.
<llm-code>
# John drives for 3 hours at a speed of 60 mph
distance_traveled_first_part = 3 * 60
# then turns around because he realizes he forgot something very important at home
distance_traveled_second_part = 0
# tries to get home in 4 hours but spends the first 2 hours in stand st ill traffic
distance_traveled_second_part += 2 * 0
# spends the next half-hour driving at a speed of 30 mph
distance_traveled_second_part += 0.5 * 30
# before being able to drive the remaining time of the 4 hours going at 80 mph
distance_traveled_second_part += 2 * 80
# total distance traveled
total_distance_traveled = distance_traveled_first_part + distance_traveled_second_part
total_distance_traveled
</llm-code>
<llm-code-output>
355.0
</llm-code-output>
Thus John is \boxed{355} miles from home at the end of those 4 hours.
```

## H   Exploring Correlation Between Q-values and Reward Scores

In this section, we explore the correlation between our learned Q-values and reward scores for partial completions, as illustrated in Table 12, and empirically investigate the possibility of reward hacking during PPO training. The partial completions are from the base model on the GSM8K test split [16]. The correlation is expected since, by design, the reward model outputs are dependent on the learned features that approximate the Q-function. This presents a potential risk in practice that the policy model may exploit during PPO. We explore attempting to decouple the reward score from the Q-value for partial completions by projecting the hidden features which approximate the Q-value through a multilayer perceptron. Table 12 and Table 13 compares the result of projecting the hidden features via a MLP layer versus a linear layer. The finding is we can reduce correlation between rewards and Q-values for partial completions while maintaining high reward model performance. Iterative retraining of the reward model on on-policy completions would further mitigate any potential reward model hacking given that our complete loss function also considers the preference ranking paired loss in 1.

| Method | IID | | OOD | | | | |
|---|---|---|---|---|---|---|---|
| | GSM8k | MATH | algebra222 | GSM-Hard | Asdiv | mawps | svamp |
| Pref. Rank | 0.805 | 0.618 | 0.850 | 0.768 | 0.829 | 0.775 | 0.761 |
| Q-Function (Linear) | **0.852** | **0.708** | **0.879** | **0.781** | 0.854 | **0.819** | 0.841 |
| Q-Function (MLP) | 0.832 | 0.693 | 0.871 | 0.770 | **0.871** | 0.780 | **0.841** |

Table 12: Comparison of AUROC scores for different reward projection strategy on several popular math benchmarks. Both Q-Function (Linear) and Q-Function (MLP) are trained with the method described in Section 3.2, the only difference being the reward head is a single linear layer versus a 3-layer MLP followed by a linear layer.

| Model | Percentile | | | |
|---|---|---|---|---|
| | 0.2 | 0.4 | 0.6 | 0.8 |
| Q-Function (Linear) | 0.983 | 0.986 | 0.988 | 0.988 |
| Q-Function (MLP) | 0.870 | 0.908 | 0.921 | 0.915 |

Table 13: Correlation between Q-value and reward score at different percentile of model generations. We see that the Q-value and reward scores for Q-Function (Linear) has higher correlation than Q-Function (MLP).

To empirically evaluate the risk of the policy gaming the reward function, we conduct a new experiment where we run PPO training for an extended time (5 episodes) in order to evaluate whether reward hacking occurs in practice. The reward scores and response lengths are shown in Figure 7 and the performance of the model at the end of each episode is given in Table 14.

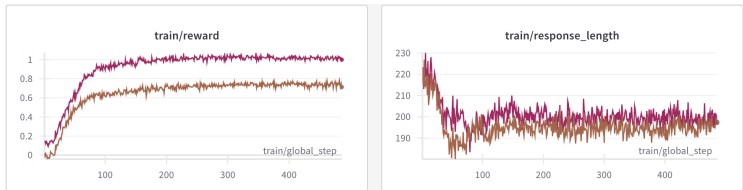

Figure 7: Train time average rewards and average response length for PPO using Codellama RM (purple) and PPO using our Q-Function RM (brown)

These results indicate training is fairly stable and the policy model does not game the reward function in practice. One explanation for why gaming does not occur is that typically during RLHF a relatively low learning rate is used, making large shifts in the models generations less likely to occur.

| Model | Episode | GSM8k | MATH | algebra222 | GSM-Hard | Asdiv | mawps | svamp |
|---|---|---|---|---|---|---|---|---|
| | 1 | 79.7 | 43.6 | 66.3 | 61.2 | 77.9 | 92.0 | 78.5 |
| | 2 | 79.6 | 45.0 | 72.5 | 61.5 | 78.0 | 91.6 | 79.9 |
| Codellama | 3 | 80.5 | 45.3 | 69.8 | 62.0 | 78.3 | 92.3 | 80.9 |
| | 4 | 80.5 | 45.5 | 70.7 | 62.5 | 78.3 | 92.0 | 79.5 |
| | 5 | 80.5 | 45.2 | 70.7 | 62.5 | 78.6 | 92.0 | 80.5 |
| | 1 | 80.2 | 45.9 | 67.6 | 62.0 | 79.2 | 93.9 | 81.6 |
| | 2 | 80.6 | 46.5 | 74.3 | 61.9 | 79.5 | 93.1 | 80.3 |
| Q-Function | 3 | 81.7 | 46.6 | 73.0 | 63.5 | 79.2 | 93.2 | 81.6 |
| | 4 | 81.0 | 46.5 | 73.4 | 63.2 | 79.2 | 93.4 | 81.0 |
| | 5 | 81.1 | 46.5 | 74.3 | 63.0 | 79.8 | 93.6 | 81.0 |

Table 14: PPO-trained policy model accuracy across several math benchmarks comparing using the baseline Codellama RM and our Q-Function RM for a single long 5 episode training run.

# I   Examples of Prototype Guided Decoding

**Question:** Consider this reference information delimited in """:

> There is a man who circles the perimeter with a baby in his arms unmoving. Locusts burn with the silhouettes of saints at dusk. Saints are in the cloud. We are in a dry storm. The man extends his circles pulling the baby through the cactus scrub. Look at his melting trainers in the heat, they arent what he asked for. There are black leather skids on the dry stone wall. People in black cloaks run out of the corner of your eye. A pig turns on a spit. The prairie is a terrarium for the blaze but the edge is dry of fire. It is the height of one season, bushes burn. A burnt five-year-old without eyelids turns quick cartwheels through the heat wave under the big pale sky, black and blue.

What type of poem is this text from?

---

**SFT:** This text is a prose poem.

---

**Prototype:** This text is a prose poem, which is a type of poetry that combines the elements of prose with those of poetry to create a unique and imaginative style of writing.

The guided decoding successfully produces a response that is still correct while adding complexity to the response by describing that the poetry combines elements of prose and imaginative writing styles. We find that, in general, guided decoding responses produce more complex responses without sacrificing correctness.

---

**Question:** Reference: <start of reference> If you invested 1,000 in Facebook in 2012, here's how much you'd have now Even given the stock market's recent volatility, a 1,000 investment in Facebook 10 years ago would still prove to be a good bet: You could have quadrupled your money. An initial investment in Facebook in 2012, when it made its initial public offering, would be worth slightly more than 4,600 as of Wednesday, according to CNBC calculations. The investment would have seen a lifetime total return of more than 360 percent, including price appreciation and dividend gains reinvested. The market's recent activity has taken a toll, of course: In one month, Facebook's stock dipped from 5,051. Its current portfolio value is just above 30 percent now, down from its peak of more than 40 percent. And the market keeps ebbing and flowing. In the charts below, all data splits are adjusted and gain-loss figures do not include dividends, interest, distributions or fees except on cash accounts. The portfolio value represents current holdings and the comparison charts represent current and historical prices of individual benchmarks, stocks or exchange-traded funds. Meanwhile, daily users in the United States and Canada dropped in 2017 to 184 million daily average users in the fourth quarter from 185 million in the third quarter. Facebook chief financial officer David Wehner, on the company's earnings call, said usage would "fluctuate" based on market penetration. The revenue increase despite the usage drop could be due to seasonal changes related to the advertising market. As the company points out, users in the fourth quarter accounted for 6.39 billion of Facebook's revenue, compared to the 5.03 billion users accounted for in the third quarter. Facebook has retreated slightly from its record price reached earlier this month as investors debate whether changes to the platform to combat fake news and other problems will hurt advertising growth. But the stock is still up 1 percent on the year. And some investors remain bullish about the stock. "News feed changes do not impair the power of the platform," Andy Hargreaves, an analyst at KeyBanc Capital Markets, wrote in a note to clients. "Facebook offers reach, targeting and ad-unit quality that is difficult to match, and they are the underlying drivers of its growth. As long as [it] continues to improve its effectiveness, the foundation for strong growth should remain intact, regardless of adjustments to the news feed." <end of reference> Instruction: What is a share of stock?

---

**SFT:** A share of stock is a unit of ownership in a company.

> **Prototype:** A share of stock is a unit of ownership in a company that represents the right to a proportionate share of the company's assets and profits.

In this response, the prototype response is slightly more correct as it describes the value of a share of stock. In addition, the prototype response is more complex than the SFT response.

> **Question:** What is the difference between a pound and an ounce? Please explain with examples.

> **SFT:** A: A pound is a unit of measurement for weight, while an ounce is a unit of measurement for volume.

> **Prototype:** A pound is equal to 0.45359237 kilograms, while an ounce is equal to 28.3495 grams. A: A pound is a unit of weight in the metric system, while an ounce is a unit of weight in the English system.

Here, we find that the SFT response is incorrect. The prototype response gives the correct response, but produces some additional information at the beginning of the response that does not answer the question. This can be seen as a side effect of the guided decoding process since a more complex response is not always desired.

> **Question:** What are some common misunderstandings about the causes and consequences of the 2008 global financial crisis?

> **SFT:** One is that it was caused by a lack of regulation in the financial sector, when in reality, the crisis was caused by a combination of factors, including excessive risk-taking by financial institutions, poor risk management practices, and a lack A: There are a few common misunderstandings about the causes and consequences of the 2008 global financial crisis.

> **Prototype:** Some believe that the crisis was caused by a lack of regulation or irresponsible lending practices. Some also believe that the crisis was caused by a lack of regulation or irresponsible lending practices. In reality, the crisis was the result of complex and interconnected events that took place over many years. There are many misconceptions about the 2008 financial crisis, including the belief that it was caused by a lack of regulation or irresponsible lending practices.

The prototype response is more coherent than the SFT response in this example. In the SFT response, there is a lack of coherence because the response is separated into two distinct sections, seperated by "A:". However, this issue does not exist in the prototype response. Each sentence builds off the last, for example "Some believe..." followed by "in-reality".

**Question:** Reference: <start of reference> As a casino player, it would be amazing to predict when a slot machine will next hit. You could time exactly when you play and how much to bet perfectly, maximising your winnings. The problem? If this was widely possible, no casino would exist. Theyd all go bankrup straight away. Instead, modern slot machines are designed to give determined results, that are not generated at random. It is incredibly difficult to guess what the result of a spin will be. There may be slot strategis that might improv your chances of winning. But any legitimate methods won't include being able to know when a slot is about to drop. Yet there's still a lot of talk about this online, with "tips" and "triks" on how to predict when a slot machine will next payout. We'll explain what really happens, along with how you can give yourself a slightly better chance of winning legitimatly. You cannot tell when a modrn slot machine will hit because the outcome of each spin is random. No matter how many times a machine has spun, and no matter what the outcomes of those spins were, the probability the next result remains the same. But people often believe otherwise. One famous example of why is the gambler's fallacy. The gambler's fallacy is an erroneous way of thinking about the probability of independent events. When people commit the gambler's fallacy, they think that the likelihood of a random event is impacted by the occurrence of previous events, or series of previous events. For example, believing that the result of previous spins on a slot will impact the result of present or future spins is committing the gambler's fallacy. The result is not affected by what's happened before. So if you win or lose one round, it has no impact on whether you'll win or lose the next round. <end of reference> Instruction: Rewrite the passage correcting any grammar and spelling mistakes that have been made. Also omit any false information that the passage may contain. Make sure to keep the same tone and message of the original passage.

**SFT:** Casino players believe they can predict when a slot machine will next hit, but this is not possible.

**Prototype:** You could time exactly when you play and how much to bet perfectly, maximising your winnings. The problem? If this was widely possible, no casino would exist. They'd all go bankrup straight away. Instead, modern slot machines are designed to give determined results, that are not generated at random. It is incredibly difficult to guess what the result of a spin will be. There may be slot strategis that might improv your chances of winning. As a casino player, it would be amazing to predict when a slot machine will next hit.

The prototype response gives more detail including additional reasoning: "modern slot machines are designed to give determined results". Interestingly, this response is generally incorrect, as modern slot machines will give random results that have deterministic rules (for example, no jackpot right after another jackpot). This shows that, while the prototype can help improve correctness, it can still suffer from hallucination.

## J  Assets

In Table 15, we summarize assets used in this paper along with their corresponding URLs and licenses.

| Asset Name | URL | License Type |
|---|---|---|
| Llama 3 [1] | https://github.com/meta-llama/llama3 | Llama 2 Community License |
| OpenMath CodeLlama [62] | https://huggingface.co/nvidia/OpenMath-CodeLlama-7b-Python | Llama 2 Community License |
| OpenRLHF [29] | https://github.com/OpenLLMAI/OpenRLHF | Apache-2.0 |
| NeMo-Skills [62] | https://github.com/Kipok/NeMo-Skills | Apache-2.0 |
| OpenMathInstruct-1 [62] | https://huggingface.co/datasets/nvidia/OpenMathInstruct-1 | NVIDIA License |
| GSM8k [16] | https://huggingface.co/datasets/gsm8k | MIT License |
| MATH [28] | https://github.com/hendrycks/math | MIT License |
| algebra222 [27] | https://huggingface.co/datasets/sirdug/Algebra222 | Apache-2.0 |
| GSM-Hard [23] | https://huggingface.co/datasets/reasoning-machines/gsm-hard | MIT License |
| Asdiv [46] | https://github.com/chaochun/nlu-asdiv-dataset | CC BY-NC 4.0 |
| mawps [35] | https://huggingface.co/datasets/MU-NLPC/Calc-mawps | MIT License |
| svamp [52] | https://huggingface.co/datasets/ChilleD/SVAMP | MIT License |
| Helpfulness and Harmlessness [8] | https://huggingface.co/datasets/Anthropic/hh-rlhf | MIT License |
| HelpSteer [67] | https://huggingface.co/datasets/nvidia/HelpSteer | CC-BY-4.0 |

Table 15: Asset information and licences.

