# OpenReview forum: "Learning Goal-Conditioned Representations for Language Reward Models"
_NeurIPS.cc/2024/Conference — NeurIPS 2024 poster_

### Official Review · Reviewer_o9bF · 2024-06-29

**Soundness:** 4
**Presentation:** 4
**Contribution:** 4
**Rating:** 8
**Confidence:** 4

**Summary:**

The paper introduces an additional contrastive loss term for reward model training that targets the learning of goal-conditioned representations that encode expected reward for partially complete sequences. The results show that this has a positive impact on reward model accuracy, downstream RLHF with the reward model, and guided generation.

**Strengths:**

The paper shows novelty and insight in using representation learning and goal conditioning to improve reward models and experimentally demonstrates the utility of doing so.

Experiments cover a number of useful metrics (spanning reward model accuracy, downstream utility in RL, downstream utility in guided generation) on appropriate benchmarks in two distinct settings (reasoning and alignment).

The paper is clearly written, with methods, results and analysis effectively communicated.

**Weaknesses:**

Whilst the paper demonstrates positive results on guided generation, the mechanism for getting the prototype seems quite arbitrary and it would be useful to present a few (maybe use case dependent) alternative approaches.

As the authors mention, the performance gains from using their reward model are less than the gains in reward model accuracy (and relatively small in general). Given the authors claim that this is most likely due to off-policy issues, it is unclear why the experiment of updating the reward model and training using the updated reward model was not run.

**Questions:**

In the appendix, the authors claim statistical significance by simply citing the size of the evaluation set. Given that confidence intervals are not reported (presumably due to computational constraints), is this claim grounded in anything more rigorous?

**Limitations:**

The authors effectively outline the limitations of their work.

---

> ### Author Rebuttal · Authors · 2024-08-07
>
> We thank the reviewer for their useful comments and insightful feedback.
> We respond to the reviewers comments and questions.
> We plan on incorporating all these responses in the final work.
>
> **"..the mechanism for getting the prototype.."** Please see our shared response to the reviewers.
> Additionally, we consider two other methods: (a) prompting the model to generate the prototype, and (b) using an auxiliary dataset (in our case, HelpSteer to construct the prototype). Both of these methods underperform the original method on Helpful-Harmless (a: $70.5 \rightarrow 69.5$, b: $70.5 \rightarrow 69.6$), asserting our original mechanism of getting the prototype.
>
> **"..updating the reward model and training.."** Thank you for your comment.
> While we state it could be possible to improve performance of the policy further by continuing to update the reward model and perform PPO (4.1.4), we choose to limit policy training in our experiments to a single iteration due to the significant labeling and computational costs associated with experimenting with further iterations of RLHF training.
>
> **"..statistical significance by simply citing the size of the evaluation set.."** Thank you for pointing this out.
> For the natural language alignment experiments, we evaluated statistical significance by performing a Student's t-test. For both the experiments in Section 4.2.2 the p-values are significant, namely, the p-value for Llama 8b Reward experiment is 0.002 and the p-value for the Q-function 8B reward experiment is 0.001.

---

> > ### Comment · Reviewer_o9bF · 2024-08-08
> >
> > I thank the authors for their further engagement and clarifications. I have read the rebuttal and maintain my score, as I see this as technically robust work with compelling results and novel insights.

---

> > > ### Author Response · Authors · 2024-08-11
> > >
> > > We thank the reviewer for their response. We are pleased to know they found the work robust and to have compelling results.

---

### Official Review · Reviewer_2p1n · 2024-07-03

**Soundness:** 4
**Presentation:** 4
**Contribution:** 3
**Rating:** 7
**Confidence:** 5

**Summary:**

The paper frames the reward learning problem for LLM as a goal-conditioned RL. It uses the contrastive learning loss from Eysenbach et al. 2022 as an additional objective for reward training. The main innovation is the adjustment of goal-conditioned RL to pairwise preference datasets. The paper shows that adding that contrastive loss on the hidden representation of the reward model can lead to a better reward model and, consequently, better policies.

**Strengths:**

* Although the proposed method is not new and relies heavily on Eysenbach et al. 2022, it has not been used before with LLMs.
* The methods and experiments are described in a straightforward and easy-to-follow manner.
* I've found the results of experiments 4.1.2 and 4.2.1 particularly interesting. There has been no change to the objective of RLHF reward models since its inception, and the proposed loss seems to be able to improve the reward model without any further annotations.

**Weaknesses:**

* The fact that the definition of a "goal state" in the language space is ambiguous since the state includes the entire generated response is ignored in the paper. Even in tasks that we have a clear "goal" in, like math, it is a bit weird. Given two responses with the same final solution but different intermediate reasoning, do you expect their hidden representation to be exactly the same? What is the meaning of averaging representations of different preferred responses? Is it supposed to be an approximation to the average cosine similarity to all vectors?
* This leads me to the fact that the prototypes used for Q-value estimation during inference time seemed a bit arbitrary. Did you ablate this during your work on the paper?

**Questions:**

* Regarding the evaluation of the reward model using the AUROC metric, can you elaborate on how you calculated it? Do you use the BT model output as classifier prediction? Where do the GT annotations come from? I looked at the references you provided (line 206), but it doesn't seem like AUROC was used there.
* In the experiment described in section 4.1.3, was the filtration done using the Q-value of the full answer or the partial one? In addition, this experiment is missing a baseline of best-of-50 using the vanilla reward model and your own reward model. This is a standard baseline when improving decoding using reward functions.
* For experiment 4.1.4, can you provide CI over multiple seeds of PPO training? It is a common practice since RL training is known to be unstable, and the performance can vary between experiments [1]. I also agree with the authors that seeing the results of an on-policy reward model will be interesting, although this can be expensive to train because of the need for annotations.
* Regarding experiment 4.2.2, it is well established that using Q values during decoding can improve performance. Wouldn't it be more relevant to compare this with other methods that use Q functions during decoding [2]? A beam search over SFT seems to me to be too weak of a baseline.

[1] Agarwal, Rishabh, et al. "Deep reinforcement learning at the edge of the statistical precipice." Advances in neural information processing systems 34 (2021): 29304-29320.

[2] Han, Seungwook, et al. "Value Augmented Sampling for Language Model Alignment and Personalization." arXiv preprint arXiv:2405.06639 (2024).

**Limitations:**

The paper properly addresses its limitations.

---

> ### Author Rebuttal · Authors · 2024-08-07
>
> We appreciate the reviewers insightful and useful comments. In the following sections, we address the questions and comments raised by the reviewer. We plan on incorporating all our discussions into the final version of this paper.
>
> **"The definition of a "goal state" in the language space..."** One of the benefits of our method is that it allows capturing goals which depend on previous information within a generation.
> For example, for reasoning tasks such as GSM8k and MATH, correct solutions depend on the final answer and intermediate reasoning.
> Similarly, for natural language tasks like helpfulness and harmfulness, a preferred solution with respect to a particular goal may depend on several parts of a generation.
> By taking a set of responses and averaging, we are able to produce a more accurate representation of the goal (see the shared reviewer response where we find averaging on more examples leads to better performance).
>
> **"...Given two responses with the same final solution but different intermediate reasoning, do you expect their hidden representation to be exactly the same?"** We observe that the representations for correct completions to the same prompt can be different depending on the reasoning path.
> For example, we run UMAP on 5K sample completions from the train set as well as a test set (GSM8k) and plot the resulting 2D plot for 7 prompts that have multiple solutions in Figure 3 within the additional supplementary material page.
>
> Still, we observe preferred and disprefferred completions are separated. In Figure 3, we also plot 5K preferred and 5K dispreffered base model completions from the train and test (GSM8k) datasets and plot the UMAP on the hidden representation.
>
> Finally, we would like to emphasize we take the average across many preferred completions, to more accurately capture the concept they represent.
> As demonstrated in the results provided in the shared reviewer response, averaging across more preferred completions produces a better representation, which improves performance.
>
> **"..What is the meaning of averaging representations of different preferred responses?"** We refer the reviewer to the shared response.
>
> **"..the prototypes used for Q-value estimation during inference time seemed a bit arbitrary. Did you ablate this during your work on the paper?"** We refer the reviewer to the shared response.
>
> **"..evaluation of the reward model using the AUROC metric, can you elaborate on how you calculated it.."**  For each problem in the benchmarks, we take the greedy generation of the base model along with annotations of whether each completion provides the correct answer, which are provided by Toshiniwal et al. [6], to compute the AUROC score.
> Concretely, for each completion, we first format using the Nemo format template and use the Reward Model to predict a reward score.
> In predicting the reward score, we also retrieve the predicted reward score for each token in the completion, not including the prompt tokens. Utilizing the predicted reward score and the annotation of the correctness of each completion, we compute an AUROC score using the Python scikit-learn package. Using the same procedure, we compute the partial AUROC scores at every tenth percentile of each completion.
>
> **"..was the filtration done using the Q-value of the full answer or the partial one.."** By definition, the Q-value is computed for each token in the sequence by taking the cosine similarity of the goal-state and the RM representation of the token, which is given by the last hidden layer. A model completion is filtered if the Q-value for any token in the completion sequence is less than the threshold, which was 0 for our experiments.
>
> **"..baseline of best-of-50 using the vanilla reward model and your own reward model.."** Since we have a total of 50 generations per problem, we use the baseline reward model (Codellama RM) and our reward model (Q-Function RM) to select the Top 1, 5, 10, and 25 samples as ranked by the reward scores.
> With the selected sample, we perform majority vote and also note the average proportion of the sample K that are correct solutions.
> The results are provided below.
> | Model| Top-K | GSM8k Accuracy (%) | GSM8k Prop. Correct (%) | MATH Accuracy (%) | MATH Prop. Correct (%) |
> |-|-|-|-|-|-|
> | Q-Function RM | 1| 84.6| 84.6| 51.7| 51.7|
> || 5| 86.2| 84.5| 59.7| 51.1|
> || 10| 86.2| 83.8| 59.5| 50.3 |
> || 25| 85.5| 81.8| 57.8| 47.0|
> |Codellama RM| 1| 80.8| 80.8| 43.8| 43.8|
> || 5| 84.3| 81.9| 54.0| 45.8|
> || 10| 85.2| 82.1| 56.0| 46.9|
> || 25| 85.8| 81.1| 56.5| 46.1|
>
> The results show that our Q-Function RM clearly outperforms the baseline.
>
> **"..CI over multiple seeds of PPO training.."** Thank you for pointing this out.
> These results are the average accuracy across 4 independent runs.
> We included the 95\% CI in Table 7 in the appendix (the CI's between the baseline and Q-Function are non-overlapping), and apologize for not including them in the main text.
> We will include them in the main text in the final version of our paper.
>
> "..other methods that use Q functions during decoding.." While there are other methods that use Q functions during decoding (Seungwook, et. al.), these rely on training a reward model (RM) and subsequently training value networks from large offline datasets (30K-100K examples).
> Our work focuses on improving the representations of RMs, so that we can compute Q values in a extremely lightweight manner.
> For instance, in 4.2.2, we use 20 examples.
> Hence, these experiments focuses on comparing with baselines in this low data setting.
>
> **"..on policy reward model.."**
> We chose to limit policy training in our experiments to a single iteration due to the significant labeling and computational costs associated with experimenting with further RLHF iterations.
>
> [6] Toshniwal, Shubham et al. “OpenMathInstruct-1: A 1.8 Million Math Instruction Tuning Dataset.” ArXiv abs/2402.10176 (2024).

---

> > ### Comment · Reviewer_2p1n · 2024-08-07
> >
> > I have read the rebuttal and found it compelling. Specifically, the best-of-N results are strong and prove that the Q-function RM is better than the vanilla one. Therefore, I've raised my score. I'll add that the clarifications regarding the choice of goal state and experimental assumptions (AUROC, CI, comparison to other methods that use Q values in decoding, etc.) improve my understanding of the paper and should be incorporated into the final version.

---

> ### Author Response · Authors · 2024-08-11
>
> We thank the reviewer for their response. We will incorporate these clarifications into the final version.

---

### Official Review · Reviewer_twr2 · 2024-07-12

**Soundness:** 3
**Presentation:** 2
**Contribution:** 2
**Rating:** 6
**Confidence:** 3

**Summary:**

This work combines contrastive representation learning, goal-conditioned RL, and reward models used in RLHF for language model alignment. The authors introduce a new method that uses a contrastive loss to encourage the reward model to learn what they define as "goal-conditioned representations." These representations essentially encode the expected reward at different steps of the generated text, supposedly helping the model predict which generations lead to good (or bad) outcomes. They test this out on math problem-solving tasks and a helpfulness/harmlessness dataset. They find some improvements in reward model accuracy and show that these representations can be used for things like filtering out bad solutions and steering the language model during generation.

**Strengths:**

- Originality: The paper creatively combines ideas from goal-conditioned RL techniques to improving reward models for LM alignment. This results in a novel method for training reward models which further improves expressiveness of the reward model via the additional loss.
- Significance: In addition to improving the overall reward models, the authors obtain an additional quantity that measures the $Q$ values of a given state (or more precisely state-action-goal), which enable a wide set of downstream use-cases, such as early misalignment detection. While these are preliminary results, they demonstrate exciting potential for further exploration.

**Weaknesses:**

- Correlation between Q-values and reward scores: The authors acknowledge the high correlation between Q-values and reward scores for partial sequences, raising concerns about the policy model potentially gaming the reward model during RLHF. While they suggest further exploration of decoupling these signals, this issue deserves more thorough analysis.
- Clarity: The paper lacks clarity in multiple areas. In particular, the functional form of the $Q$ function is not entirely clear to me. Choosing the goal states is another source of obscurity. While it is clear that during training, positive and negative examples are chosen from the respective completions, in my opinion the more important part, which is the choice of goal states during inference is not entirely clear to me. The authors briefly mention that they "we ake the mean representation of multiple goal state completions from the training data", but given that this is an important part of the work, I believe it requires a lot more analysis.

**Questions:**

- Have you considered investigating approaches such as Upside Down RL [1, 2] given that conditioning in this case is not on a goal state but the desired rating?

1. https://arxiv.org/abs/1912.02875
2. https://arxiv.org/abs/2106.01345

**Limitations:**

The authors have adequately addressed the limitations of their work.

---

> ### Author Rebuttal · Authors · 2024-08-07
>
> We thank the reviewer for useful comments and for appreciating the originality and performance improvements of the method.
> We address the reviewers questions and comments in the following sections.
> Additionally, we plan on incorporating all our responses and discussions into the final version of the paper.
>
> **Correlation between Q-values and reward scores** Thank you for your comment.
> To evaluate the risk of the policy gaming the reward function, we conduct a new experiment where we run PPO training for an extended time (5 episodes) in order to evaluate whether reward hacking occurs in practice.
> The reward scores and response lengths are shown in Figure 2 within the additional supplementary
> material page and the performance of the model at the end of each episode is given here.
>
> | Model        | Episode | GSM8k | MATH | algebra222 | GSM-Hard | Asdiv | mawps | svamp |
> |--------------|---------|-------|------|------------|----------|-------|-------|-------|
> | Codellama    | 1       | 79.7  | 43.6 | 66.3       | 61.2     | 77.9  | 92.0  | 78.5  |
> |              | 2       | 79.6  | 45.0 | 72.5       | 61.5     | 78.0  | 91.6  | 79.9  |
> |              | 3       | 80.5  | 45.3 | 69.8       | 62.0     | 78.3  | 92.3  | 80.9  |
> |              | 4       | 80.5  | 45.5 | 70.7       | 62.5     | 78.3  | 92.0  | 79.5  |
> |              | 5       | 80.5  | 45.2 | 70.7       | 62.5     | 78.6  | 92.0  | 80.5  |
> | Q-Function   | 1       | 80.2  | 45.9 | 67.6       | 62.0     | 79.2  | 93.9  | 81.6  |
> |              | 2       | 80.6  | 46.5 | 74.3       | 61.9     | 79.5  | 93.1  | 80.3  |
> |              | 3       | 81.7  | 46.6 | 73.0       | 63.5     | 79.2  | 93.2  | 81.6  |
> |              | 4       | 81.0  | 46.5 | 73.4       | 63.2     | 79.2  | 93.4  | 81.0  |
> |              | 5       | 81.1  | 46.5 | 74.3       | 63.0     | 79.8  | 93.6  | 81.0  |
>
> These results indicate training is fairly stable and the policy model does not game the reward function in practice.
> One explanation for why gaming does not occur is that typically during RLHF a relatively low learning rate is used, making large shifts in the models generations less likely to occur.
>
> **"..the functional form of the Q function.."** The Q function is parameterized by a scoring function $f$ and encoder.
> In this paper, we use the cosine similarlity as $f$ and the encoder is a causal LM, such as Llama.
> In particular, to compute the Q-value for a goal-state and state-action pair, we embed both the goal state and the hidden state for the state-action pair, and take the cosine similarity between the two representations.
>
>
> **"..choice of goal states during inference"** We refer the reviewer to the shared response.
>
> **"..investigating approaches such as Upside Down RL.."** Our work focuses on improving the representations learned by reward models (RMs) for aligning LMs.
> Our improved methods of training RMs leads to improvements in RM performance, downstream policy learning, and model steerability.
> Approaches such as upside down RL do not focus on improving RM performance to better align LMs, so we did not consider investigating them in this work.

---

> > ### Comment · Reviewer_twr2 · 2024-08-08
> > **Further clarification welcome**
> >
> > Thank you for providing detailed clarifications. After reading your rebuttal and the other reviews, and after going through the paper again, I can say that I am happy with the clarification and your ablations on the goal state used. Before making updates, I would like to ask authors for two more clarifications that came up during this process.
> >
> > 1. When selecting the goal state during inference, you assume a set of completions. What is this set - is it the full training dataset, or some parts of the training dataset, and does this set (and the resulting goal state) change across experiments or does it stay fixed? More concretely, in 4.1.3 you compute Q-values to filter completions - is the goal state used here the fixed goal state from the training dataset or is it constructed in a different manner?
> > 2. In 4.1.2, could you please explain in more detail the x-axis of Figure 2? Does the percentile refer to the set of generations ordered by reward?

---

> > > ### Author Response · Authors · 2024-08-08
> > >
> > > Thank you for your clarification questions.
> > >
> > > 1. We refer the reviewer to the **Computing Q-values** paragraph of the paper (line 165, Section 3.2). In particular, the set of completions used for the experiment in 4.1.3 is the unique preferred completions from the Preference Ranking Dataset. This choice stays constant in experiments, except for the steering experiments (4.2.2), where we construct the goal state from a smaller set of examples in order to evaluate steerability from a smaller set of examples.
> > >
> > >
> > > 2. In Figure 2, the percentile refers to the percent of the completion considered. For instance, if a completion has 100 tokens and the percentile is 0.2, we consider the reward score placed on the 20th token. To compute the AUROC score, we compare the reward scores assigned at the 20th percentile of all completions and the annotation of whether the completion is correct.

---

> > > > ### Comment · Reviewer_twr2 · 2024-08-12
> > > >
> > > > Thank you for the detailed follow up clarifications. After going through all the discussions I have decided to update my score and accept the paper.

---

> > > > > ### Author Response · Authors · 2024-08-12
> > > > >
> > > > > Thank you for your response!

---

### Official Review · Reviewer_U1X3 · 2024-07-13

**Soundness:** 3
**Presentation:** 2
**Contribution:** 3
**Rating:** 6
**Confidence:** 3

**Summary:**

This paper presents a method of applying goal-conditioned Q-functions to learn representations via contrastive learning to capture the expected reward. By incorporating an auxiliary contrastive loss for training the reward model, the performance of language model alignment obtains improvement. Experiments on GSM8k and MATH further validate the superior performance of the proposed method.

**Strengths:**

- This paper adapts the goal-conditioned representation learning in RL to help boost the performance of the language-based reward model for LLM alignment, which is novel in the LLM area.
- A thorough set of experiments demonstrates the superior performance of the proposed method.

**Weaknesses:**

See questions.

**Questions:**

- How is the performance of the proposed method compared with DPO? Since DPO is a well-known LLM alignment method, I encourage the authors to add a comparison and discussion between them.

**Limitations:**

Limitations have been discussed in this paper.

---

> ### Author Rebuttal · Authors · 2024-08-07
>
> We are grateful to the reviewer for their thoughtful review and insightful suggestions.
> We are pleased to know they acknowledge the novelty of the goal-conditioned approach with LLMs and found our experiments to be thorough.
> The reviewer brings up the interesting point of comparing our proposed method with DPO.
> We compare the performance of DPO with our proposed method on the mathematical reasoning with code execution tasks (Section 4.1).
> Furthermore, we intend to incorporate this response and discussion into the final version of the paper.
>
> In particular, we use the GSM8K + MATH preference ranking dataset that we used for training both the baseline and contrastive reward models as the DPO training dataset.
> We compare performance of DPO with PPO training using the baseline reward model and our contrastive reward model.
> We present average accuracy across 4 independent runs for PPO and 2 independent runs for DPO.
> The base model results are also shown as a reference point presented in [6].
>
> | Model           | GSM8k             | MATH              | algebra222        | GSM-Hard          | Asdiv             | mawps             | svamp             |
> |-----------------|-------------------|-------------------|-------------------|-------------------|-------------------|-------------------|-------------------|
> | Base            | 75.9              | 43.6              | 65.6              | 60.1              | 77.7              | 93.5              | 79.6              |
> | DPO             | 80.1 ± 0.1        | 44.8 ± 0.2        | 64.8 ± 0.8        | 59.9 ± 0.4        | 76.9 ± 0.0        | 90.3 ± 0.2        | 76.6 ± 0.1        |
> | Codellama PPO   | 79.3 ± 0.2        | 43.4 ± 0.2        | 65.8 ± 1.6        | 61.1 ± 0.3        | 77.4 ± 0.2        | 91.6 ± 0.3        | 78.5 ± 0.9        |
> | Q-Function PPO  | **80.5 ± 0.3**    | **45.1 ± 0.1**    | **70.9 ± 1.7**    | **62.7 ± 0.5**    | **79.5 ± 0.5**    | **93.6 ± 0.3**    | **81.2 ± 0.4**    |
>
> Interestingly, we observe that DPO performs on par with our method for the In-Distribution tasks (GSM8k and MATH). However, the policy trained with the contrastive RM performs much better on OOD tasks (algebra222, GSM-Hard, Asdiv, mawps, and svamp), compared to DPO.
> These findings could be explained by related works that analyze DPO and find it can perform poorly in OOD settings [1]. All in all, these results indicate that PPO training with the contrastive RM leads to better generalization than training with DPO, particularly in OOD settings.
>
> [1] Xu, Shusheng et al. “Is DPO Superior to PPO for LLM Alignment? A Comprehensive Study.” ArXiv abs/2404.10719 (2024).
>
> [6] Toshniwal, Shubham et al. “OpenMathInstruct-1: A 1.8 Million Math Instruction Tuning Dataset.” ArXiv abs/2402.10176 (2024).

---

> > ### Comment · Reviewer_U1X3 · 2024-08-08
> >
> > Thanks for your response and hard work. After reading your rebuttal, I've decided to maintain my score. I tend to accept this paper for its novelty and good performance.

---

> > > ### Author Response · Authors · 2024-08-11
> > >
> > > Thank you for your response. We are pleased to hear the reviewer found our work novel and to have good performance.

---

### Author Rebuttal · Authors · 2024-08-07

We thank the reviewers for their valuable and insightful comments.
It is appreciated that the reviewers found our work to be well-written, novel, and insightful for improving RM and policy performance.
In this section, we provide further elaboration on our method of constructing the goal state.
We will include this discussion and results in the final paper.

**Goal State Construction**

Because reviewers brought up several questions about the goal state, we briefly review how we construct the goal state.
In order to construct the goal state, given a set of completions $\\{y_i = [y_{(i, 0)}, \dots, y_{(i, t_i)}] | i = 1, \dots, N\\}$ we take the last reward model hidden state of the final token, $h(y_{(i, t_i)})$, and then we average across all completions: $ \frac{1}{N} \sum_{i=1}^N h(y_{(i, t_i)}) $, where $h$ represents the last hidden layer of the reward model.

**Choice of Goal State**

Reviewers brought up several questions about why we chose to average across a set of preferred completions from the training dataset as the goal state during inference.
We experimented with using the mean across a set of completions that contain certain desirable attributes (e.g., being correct or helpful) as the goal state.
Our choice of taking the mean across these representations is motivated by prior work, which has found taking averages across representations can more accurately encode concepts or relationships [2, 3].
For example, by averaging across the set of correct completions in Math, we better capture the notion of the solution being correct.

To ablate this choice of goal state during inference, we conduct a new experiment under the mathematical reasoning with code setting (Section 4.1) where we evaluate the effect of (a) choosing poor examples to construct the goal state and (b) the number of examples used to construct the goal state.
In particular, we evaluate 3 settings for picking the sample completions for constructing the goal state.
First, we vary the number of preferred completions used to construct the goal state to evaluate the effect of adding more examples of good reasoning.
Second, we vary the number of dispreferred completions used to construct the goal state, to evaluate whether bad examples lead to poor performance.
Third, we incrementally add more dispreferred completions to a fixed sample of preferred completion in order to measure the robustness of our goal state computation to negative examples.
We refer to the addition of dispreferred completions on top of all preferred completions as adding ``corrupted'' examples.
The results are in Figure 1, within the additional supplementary material page.
Overall, these results show that negative or unhelpful completions degrade performance, indicating the importance of choosing relevant examples for computing the goal state.
Additionally, they demonstrate that having more examples of the concept leads to better performance as more generations are filtered with comparable proportion of those remaining generations being correct.

We additionally ablate our choice of using the last token in the completion sequence to construct the goal state.
In this experiment, we repeat our filtering experiments from 4.1.3.
Except, we randomly sample a token from in the completion sequence and use it as the goal state:

| Sampling Method | GSM8k Accuracy (%) | GSM8k Prop. Correct (%) | MATH Accuracy (%) | MATH Prop. Correct (%) |
|-----------------|--------------------|-------------------------|-------------------|------------------------|
| Last Token      | 86.0               | 84.0                    | 59.6              | 52.0                   |
| Random Token    | 85.9               | 81.8                    | 57.3              | 45.3                   |


From the result of this ablation, we see that last token sampling has superior performance, indicating using the last token helps achieve better performance.


[2] Mikolov, Tomas et al. “Efficient Estimation of Word Representations in Vector Space.” International Conference on Learning Representations (2013).

[3] Le, Quoc V. and Tomas Mikolov. “Distributed Representations of Sentences and Documents.” International Conference on Machine Learning (2014).

---

### Decision · Program_Chairs · 2024-09-25

**Decision:**

Accept (poster)

**Comment:**

This paper proposes a method to bias reward models in RLHF to align with goals at intermediate stages of the input sequence, using contrastive learning. This helps using these rewards to prune outputs during the generation process and to even improve on some benchmarks. Reviewers generally appreciated the paper’s presentation and contributions. Certain discussion points (e.g., averaging over completions, comparison with DPO in-distribution and OOD, clarifying the Q function, base-of-k baseline) are worth including in the paper.